# Modality-agnostic decoding of vision and language from fMRI

**Mitja Nikolaus[1][*][†], Milad Mozafari[2][†], Isabelle Berry[1], Nicholas Asher[3], Leila Reddy[1], Rufin VanRullen[1]**

[1]Université de Toulouse, CNRS, CerCo, Toulouse, France; [2]Torus AI, Toulouse, France; [3]Université de Toulouse, IRIT, Toulouse, France

## eLife Assessment

The study introduces a **valuable** dataset for investigating the relationship between vision and language in the brain. The authors provide **convincing** evidence that decoders trained on brain responses to both images and captions outperform those trained on responses to a single modality. The dataset and decoder results will be of interest to communities studying brain and machine decoding.

**\*For correspondence:**
mitja.nikolaus@posteo.de

[†]These authors contributed equally to this work

**Competing interest:** The authors declare that no competing interests exist.

**Abstract** Humans perform tasks involving the manipulation of inputs regardless of how these signals are perceived by the brain, thanks to representations that are invariant to the stimulus modality. In this paper, we present modality-agnostic decoders that leverage such modality-invariant representations to predict which stimulus a subject is seeing, irrespective of the modality in which the stimulus is presented. Training these modality-agnostic decoders is made possible thanks to our new large-scale fMRI dataset SemReps-8K, released publicly along with this paper. It comprises six subjects watching both images and short text descriptions of such images, as well as the conditions during which the subjects were imagining visual scenes. We find that modality-agnostic decoders can perform as well as modality-specific decoders and even outperform them when decoding captions and mental imagery. Furthermore, a searchlight analysis revealed that large areas of the brain contain modality-invariant representations. Such areas are also particularly suitable for decoding visual scenes from the mental imagery condition.

## Introduction

Several regions in the human brain have developed a high degree of specialization for particular lower-level perceptive as well as higher-level cognitive functions (*Kanwisher, 2010*). For many higher-level functions, it is crucial to be able to manipulate inputs regardless of the modality in which a stimulus was perceived by the brain. Such manipulations can be performed thanks to representations that are abstracted away from particularities of specific modalities and are, therefore, *modality-invariant*. A range of theories have been developed to explain how and where in the human brain such invariant representations are created (*Damasio, 1989*; *Binder et al., 2009*; *Martin, 2016*; *Barsalou, 2016*; *Ralph et al., 2017*).

In this paper, we aim to develop modality-agnostic decoders that are specifically trained to leverage such modality-invariant representation patterns in the brain. In contrast to *modality-specific* decoders that are trained to be applied only in the modality that they were trained on, *modality-agnostic* decoders can be applied to decode brain patterns from multiple modalities, even without knowing a priori the modality the stimulus was presented in.

Regarding the terminology that will be used in the following, we use the terms modality-agnostic and modality-specific to refer to the way decoders are trained. The term modality-dependent describes patterns in brain activity that are completely unrelated to stimuli presented in varying modalities. In contrast, modality-invariant patterns do contain a significant degree of shared representational structure between modalities (*Man et al., 2012*). Evidence suggests that many regions in the brain contain mixtures of both types of information: A single region can show activation patterns with contributions of multiple modality-dependent and modality-invariant features (*Liuzzi et al., 2020*; *Dirani and Pylkkänen, 2024*). Modality-agnostic decoders are explicitly trained to leverage the shared information in modality-invariant patterns of the brain activity.

In order to build modality-agnostic decoders, large multimodal neuroimaging datasets with well-controlled stimuli across modalities are required. While large-scale datasets exist for vision (*Huth et al., 2012*; *Chang et al., 2019*; *Allen et al., 2022*), language (*Brennan and Hale, 2019*; *Nastase et al., 2021*; *Schoffelen et al., 2019*; *Tang et al., 2023a*), and video (naturalistic movies) (*Aliko et al., 2020*; *Visconti di Oleggio Castello et al., 2020*; *Boyle et al., 2020*), none of these contain a controlled set of equivalent stimuli that are presented separately in both modalities. For instance, the different modalities in movies (vision and language) are complementary but do not always carry the same semantics. Furthermore, they are not presented separately but simultaneously, impeding a study of the respective activity pattern caused by each modality in isolation.

The analyses of this paper are based on **SemReps-8K**, a new large-scale multimodal fMRI dataset of six subjects each viewing more than 8000 stimuli which are presented separately in one of two modalities, as images of visual scenes or as descriptive captions of such images. In addition, the dataset also contains three imagery conditions for each subject, where they had to imagine a visual scene based on a caption description they had received before the start of the fMRI experiment.

We exploit this new data to develop modality-agnostic decoders that are trained on brain imaging data from multiple modalities, which we demonstrate here for the case of vision and language. We find that modality-agnostic decoders trained on this dataset perform on par with their respective modality-specific counterparts for decoding images, despite the additional challenge of uncertainty about the stimulus modality. For decoding captions, the modality-agnostic decoders even outperform the respective modality-specific decoders (because the former, but not the latter, can leverage the additional training data from the other image modality).

Additionally, we use this novel kind of decoders for a searchlight analysis to localize regions with modality-invariant representations in the brain. Previous studies that aimed to localize modality-invariant patterns were based on limited and rather simple stimulus sets and did not always agree on the exact location and extent of such regions (e.g. *Vandenberghe et al., 1996*; *Shinkareva et al., 2011*; *Devereux et al., 2013*; *Fairhall and Caramazza, 2013*; *Jung et al., 2018*). We design a searchlight analysis based on a combination of modality-agnostic decoders and cross-decoding. The results reveal that modality-invariant patterns can be found in a widespread left-lateralized network across the brain, encompassing virtually all regions that have been proposed previously.

Finally, we find that modality-agnostic decoders trained only on data with perceptual input also generalize to conditions during which the subjects were performing mental imagery and outperform both kinds of modality-specific decoders on this task.

## Related work

### Modality-invariant representations

Decades of neuro-anatomical research (e.g. based on clinical lesions) and electrophysiology in non-human and human primates, as well as modern experiments leveraging recent brain imaging techniques have provided evidence that the activation patterns in certain brain areas are modality-dependent; for example, the occipital cortex responds predominantly to visual stimulation (*Felleman and Van Essen, 1991*; *Sereno et al., 1995*; *Grill-Spector and Malach, 2004*), and a commonly left-lateralized network responds to language processing tasks (*Zola-Morgan, 1995*; *Fedorenko et al., 2010*; *Fedorenko et al., 2011*; *Friederici, 2017*; *Brennan, 2022*).

More recent research has started to focus on higher-level regions that respond with *modality-invariant* patterns, i.e., patterns that are abstracted away from any modality-dependent information. A modality-invariant region responds with similar patterns to input stimuli of the same meaning, even if they are presented in different modalities (e.g. the word 'cat' and picture of a cat).

Several theories and frameworks on how the brain transforms inputs from different modalities into modality-invariant representations have been proposed. (Here, we do not attempt to disambiguate whether information is modality-invariant or completely abstracted away from input modalities, described in the literature as modality-independent *Dirani and Pylkkänen, 2024*, supramodal *Sanchez et al., 2020*, transmodal *Ralph et al., 2017*, amodal *Fairhall and Caramazza, 2013*, or abstract/conceptual *Binder, 2016* (see also distinctions made in *Barsalou, 2016*). For our goal of building modality-agnostic decoders, it is only important that information be represented in a way invariant to the stimulus modality. We will come back to this point in the Discussion section.) The **convergence zones view** proposes that modality-dependent information coming from sensory cortices is integrated in multiple convergence zones that are distributed across the cortex, predominantly in temporal and parietal lobes (*Damasio, 1989*; *Tranel et al., 1997*; *Meyer and Damasio, 2009*). These convergence zones are organized hierarchically, learned associations are used to create abstractions from lower-level to higher-level feature representations (*Simmons and Barsalou, 2003*; *Meyer and Damasio, 2009*). A perceived stimulus first causes modality-dependent activity in the related low-level region (e.g. visual cortex), subsequently, higher-level convergence zones serve as relays that cause associated activity in other regions of the brain (e.g. the language network) (*Meyer and Damasio, 2009*; *Kiefer and Pulvermüller, 2012*). According to *Binder, 2016*, the most high-level convergence zones can become so abstract that they are representing amodal symbols.

The **GRAPES framework** (Grounding Representations in Action, Perception, and Emotion Systems) also suggests that representations are distributed across temporal and parietal areas of the cortex. More specifically, they are hypothesized to be situated in areas connected to the perception and manipulation of the environment, as well as in the language system (*Martin, 2009*; *Martin, 2016*). According to this theory, conceptual knowledge is organized in domains: For example, semantic information related to object form and object motion is represented within specific visual processing systems, regardless of the stimulus modality, and both for perception as well as imagination.

The **hub-and-spoke theory** states that cross-modal interactions are mediated by a single modality-invariant hub, located in the anterior temporal lobes (*Rogers et al., 2004*; *Lambon Ralph et al., 2006*; *Patterson and Lambon Ralph, 2016*; *Ralph et al., 2017*). The hub contains a 'continuous distributed representation space that expresses conceptual similarities among items even though its dimensions are not independently interpretable' (*Frisby et al., 2023*, p. 262). The spokes form the links between the hubs and the modality-dependent association cortices. Most importantly, semantic representations are not solely based in the hub; for a given concept, all spokes that are linked to modalities in which the concept can be experienced do contribute to the semantic representation. This explains why selective damage to spokes can cause category-specific deficits (*Pobric et al., 2010*).

This is conceptually similar to some aspects of the **Global Workspace Theory**, which assumes both a multimodal convergence of inputs towards a specific (network of) region(s), and the possibility of flexibly recruiting unimodal regions into this Global Workspace (*Baars, 1993*; *Baars, 2005*).

While these and other theories partly disagree on *how* modality-invariant information is represented in the brain, they agree that such information is distributed across the cortex, and possibly overlapping with the semantic network (*Binder et al., 2009*; *Huth et al., 2012*; *Andrews et al., 2014*; *Zwaan, 2016*).

## Decoding of vision and language from fMRI

Early approaches to brain decoding focused on identifying and reconstructing limited sets of simple visual stimuli (*Haxby et al., 2001*; *Cox and Savoy, 2003*; *Kay et al., 2008*; *Naselaris et al., 2009*; *Nishimoto et al., 2011*). Soon after, attempts to decode linguistic stimuli could identify single words and short paragraphs with the help of models trained to predict features extracted from word embeddings (*Pereira et al., 2018*).

More recently, large-scale open-source fMRI datasets for both vision and language have become available (*Chang et al., 2019*; *Allen et al., 2022*; *Schoffelen et al., 2019*; *Tang et al., 2023b*) and allowed for the training of decoding models for a larger range and more complex naturalistic stimuli with the help of features extracted from deep learning models. For example, modality-specific decoders for vision can be trained by mapping the brain activity of subjects viewing naturalistic images to feature representation spaces of computational models of the same modality (i.e. vision models) (*Shen et al., 2019*; *Beliy et al., 2019*; *Lin et al., 2022*; *Takagi and Nishimoto, 2023*; *Ozcelik*

and VanRullen, 2023). Moreover, a range of studies provided evidence that certain representations can transfer between vision and language by evaluating decoders in a modality that they were not trained on *Shinkareva et al., 2011*; *Man et al., 2012*; *Fairhall and Caramazza, 2013*; *Simanova et al., 2014*; *Jung et al., 2018*. The performance in such *cross-modal decoding* evaluations always lags behind when compared to within-modality decoding. One explanation is that modality-specific decoders are not explicitly encouraged to pick up on modality-invariant features during training, and modality-dependent features do not transfer to other modalities (an exception is the case of modality-specific decoders that are trained using features from the other modality; such decoders might also rely on modality-invariant features, however these might be less easily decodable than the modality-dependent features).

To address this limitation, we here propose to directly train *modality-agnostic decoders*, i.e., models that are exposed to multiple stimulus modalities during training in order to make it more likely that they are leveraging representations that are modality-invariant. Training this kind of decoder is enabled by the multimodal nature of our fMRI dataset: The stimuli are taken from COCO, a multimodal dataset of images with associated descriptive captions (*Lin et al., 2014*). During the experiment, the subjects are exposed to stimuli in both modalities (images and captions) in separate trials. The modality-agnostic decoders learn to map brain activity associated with a specific stimulus independent of its presentation modality (image or caption) to high-level features extracted from a deep learning model. After training, a single modality-agnostic decoder can be used to decode stimuli from multiple modalities, leveraging representations that are invariant to the stimulus modality.

## Decoding of mental imagery

Apart from decoding perceived stimuli, it is also possible to decode representations when subjects were performing mental imagery, without being exposed to any perceptual input.

Different theories on mental imagery processes emphasize either the role of the early visual areas (*Kosslyn et al., 1999*; *Pearson, 2019*) or the role of the high-level visual areas in the ventral temporal cortex and frontoparietal networks (*Spagna et al., 2021*; *Hajhajate et al., 2022*; *Liu et al., 2025*). There is evidence for both kinds of theories in the form of neuroimaging studies that used decoding to identify stimuli during mental imagery. Some of these found relevant patterns in the early visual cortex (*Albers et al., 2013*; *Naselaris et al., 2015*), others highlighted the role of higher-level areas in the ventral visual processing stream (*Stokes et al., 2009*; *Reddy et al., 2010*; *Lee et al., 2012*; *VanRullen and Reddy, 2019*; *Boccia et al., 2019*) as well as the precuneus and the intraparietal sulcus (*Johnson and Johnson, 2014*). These discrepancies can possibly be explained by differences in experimental design: For example, the early visual cortex might only become involved if the task requires the imagination of high-resolution details, which are represented in lower levels of the visual processing hierarchy (*Kosslyn and Thompson, 2003*).

Crucially, it has been shown that decoders trained exclusively on trials *with* perceptual input can generalize to imagery trials (*Stokes et al., 2009*; *Reddy et al., 2010*; *Lee et al., 2012*; *Johnson and Johnson, 2014*; *Naselaris et al., 2015*), providing evidence that representations formed during perception overlap to some degree with representations formed during mental imagery (*Dijkstra et al., 2019*).

In our study, we explored to what extent these findings hold true for more varied and complex stimuli. Following previous approaches, we use decoders trained exclusively on trials where subjects were viewing images and captions and evaluated them on their ability to decode the imagery trials. We additionally hypothesized that mental imagery should be at least as conceptual as it is sensory and, therefore, primarily recruits modality-invariant representations. Consequently, modality-agnostic decoders should be ideally suited to decode mental imagery and outperform modality-specific decoders on that task.

## Localizing modality-invariant regions

The first evidence for the existence of modality-invariant regions came from observations of patients with lesions in particular cortical regions which led to deficits in the retrieval and use of knowledge across modalities (*Warrington and Shallice, 1984*; *Warrington and Mccarthy, 1987*; *Gainotti, 2000*; *Damasio et al., 2004*). Semantic impairments across modalities have also been observed in patients

with the neurodegenerative disorder semantic dementia (*Warrington, 1975*; *Snowden et al., 1989*; *Jefferies et al., 2009*).

In early work exploring the possible locations of modality-invariant regions in healthy subjects, brain activity was recorded using imaging techniques while they were presented with a range of concepts in two modalities (e.g. words and pictures). Regions that were active during semantic processing of stimuli in the first modality were compared to regions that were active during semantic processing of stimuli in the second modality. The conjunction of these regions was proposed to be modality-invariant (*Vandenberghe et al., 1996*; *Moore and Price, 1999*; *Bright et al., 2004*).

While this methodology allows for the identification of candidate regions in which semantic processing of multiple modalities occurs, it cannot be used to probe the information represented in these regions. In order to compare the information content (i.e. multivariate patterns) of brain regions, researchers have developed Representational Similarity Analysis (RSA, *Kriegeskorte et al., 2008*) as well as encoding and decoding analyses (*Naselaris et al., 2011*). More specifically, RSA has been used to find modality-invariant regions by comparing activation patterns of a candidate region when subjects are viewing stimuli from different modalities (*Devereux et al., 2013*; *Handjaras et al., 2016*; *Liuzzi et al., 2017*). This comparison is performed in an indirect way, by measuring the correlation of dissimilarity matrices of activation patterns. In turn, cross-decoding analysis can be leveraged to identify modality-invariant regions by training a classifier to predict the category of a stimulus in a given modality, and then evaluating its performance to predict the category of stimuli that were presented in another modality (*Shinkareva et al., 2011*; *Man et al., 2012*; *Fairhall and Caramazza, 2013*; *Simanova et al., 2014*; *Jung et al., 2018*). However, all these studies relied on a predefined set of stimulus categories and can, therefore, not be easily extended to more realistic and complex stimuli, as we perceive them in our everyday life.

We summarize candidates for modality-invariant regions that have been identified by previous studies in Appendix 4. This overview reveals substantial disagreement regarding the possible locations of modality-invariant patterns in the brain. For example, *Fairhall and Caramazza, 2013* found modality-invariant representations in the left ventral temporal cortex (fusiform, parahippocampal, and perirhinal cortex), middle and inferior temporal gyrus, angular gyrus, parts of the prefrontal cortex as well as the precuneus. *Shinkareva et al., 2011* found a larger network of left-lateralized regions, including additionally the left superior temporal, inferior parietal, supramarginal, inferior and inferior occipital, precentral and postcentral gyrus, supplementary motor area, intraparietal sulcus, cuneus, posterior cingulum as well as the right fusiform gyrus and the superior parietal gyrus, paracentral lobule on both hemispheres. In contrast, *Jung et al., 2018* found modality-invariant representations only in the right prefrontal cortex. These diverging results can probably be explained by the limited number of stimuli as well as the use of artificially constructed stimuli in certain studies.

Recent advances in machine learning have enabled another generation of fMRI analyses based on large-scale naturalistic datasets. Here, we present a new multimodal dataset of subjects viewing both images and text. Most importantly, the dataset contains a large number of naturalistic stimuli in the form of complex visual scenes and full sentence descriptions of the same type of complex scenes, instead of pictures of single objects and words as commonly used in previous studies. This data enables the development of modality-agnostic decoders that are explicitly trained to leverage features that are shared across modalities. Furthermore, we use this data to localize modality-invariant regions in the brain by applying decoders in a multimodal searchlight analysis.

## Methods
### fMRI experiment

Six subjects (three female, age between 22 and 47 years, all right-handed and fluent English speakers) participated in the experiment after providing informed consent. The study was performed in accordance with French national ethical regulations (Comité de Protection des Personnes, ID 2019-A01920-57). We collected functional MRI data using a 3T Philips ACHIEVA scanner (gradient echo pulse sequence, TR = 2 s, TE = 30 ms, 46 slices with a 32-channel head coil, slice thickness = 3 mm with 0.2 mm gap, in-plane voxel dimensions 3×3 mm). At the start of each session, we further acquired high-resolution anatomical images for each subject (voxel size = 1 mm$^3$, TR = 8.13 ms, TE = 3.74 ms, 170 sagittal slices).

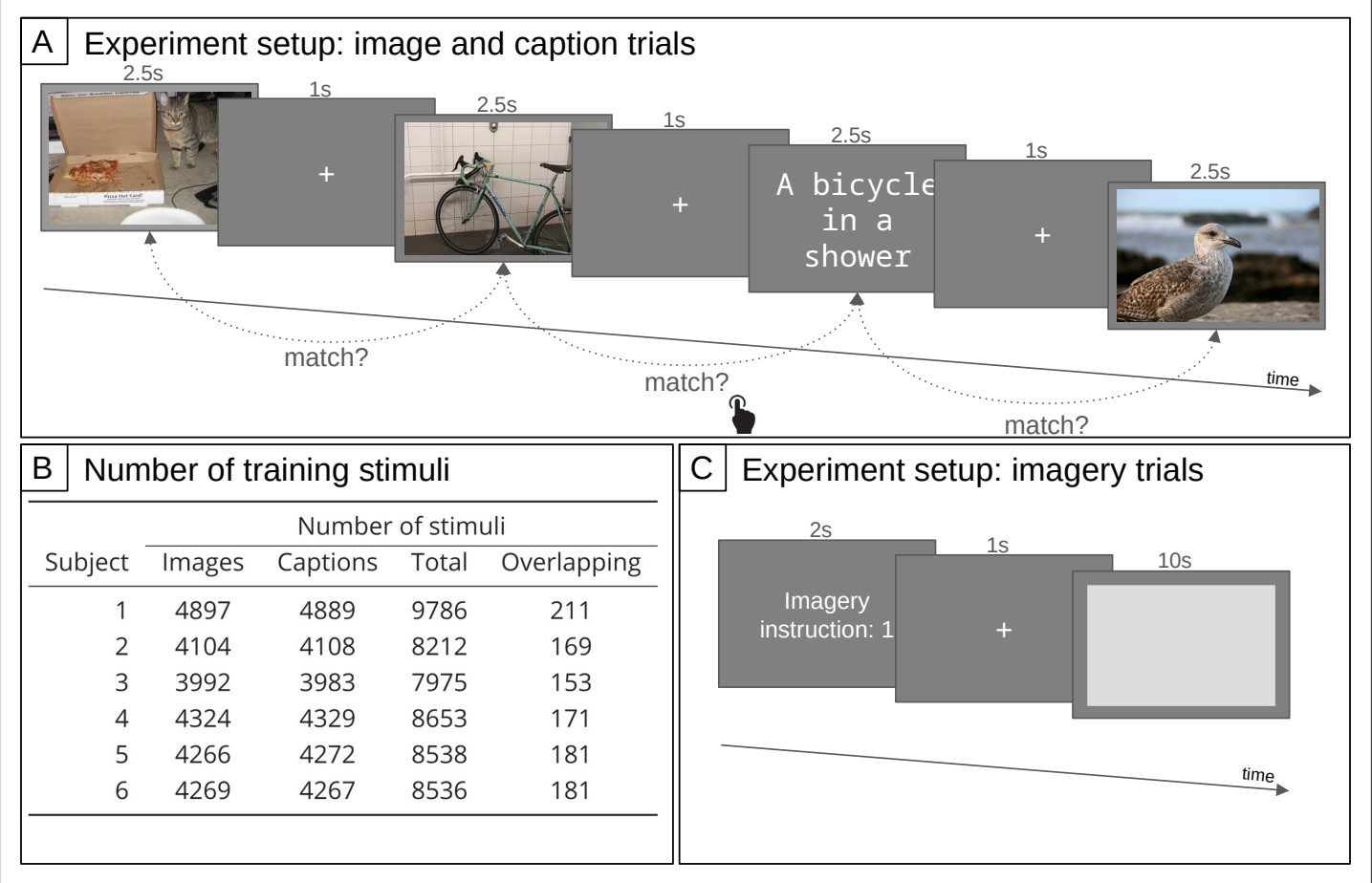

**Figure 1.** Setup of the fMRI experiment. (**A**) Setup of the main fMRI experiment. Subjects were seeing images and captions in random alternation. Whenever the current stimulus matched the previous stimulus, the subjects were instructed to press a button (one-back matching task). Images and captions for illustration only; actual size and stimuli as described in the text. (**B**) Number of distinct training stimuli (excluding trials that were one-back targets or during which the subject pressed the response button). The number of overlapping stimuli indicates how many stimuli were presented both as caption and as image. There was an additional set of 140 stimuli (70 images and 70 captions) used for testing. (**C**) Setup of the fMRI experiment for the imagery trials. Subjects were instructed to remember three image descriptions with corresponding indices (numbers 1–3). One of these indices was displayed during the instruction phase, followed by a fixation phase, and then the subjects were imagining the visual scene for 10 s.

Scanning was spanned over 10 sessions (except for sub-01: 11 sessions), each consisting of 13–16 runs during which the subjects were presented with 86 stimuli. Each run started and ended with an 8 s fixation period. The stimulus type varied randomly inside each run between images and captions. Each stimulus was presented for 2.5 s at the center of the screen visual angle: 14.6 degrees; captions were displayed in white on a dark gray background (font: 'Consolas'), the inter-stimulus interval was 1 s. Every 10 stimuli, there was a fixation trial that lasted for 2.5 s. Every 5 min there was a longer fixation trial for 16 s.

Subjects performed a one-back matching task: They were instructed to press a button whenever the stimulus matched the immediately preceding one (**Figure 1** Panel A). In case the previous stimulus was of the same modality (e.g. two captions in a row), the subjects were instructed to press a button if the stimuli matched exactly. In the cross-modal case (e.g. an image followed by a caption), the button had to be pressed if the caption was a valid description of the image, and vice versa. Positive one-back trials occurred on average every 10 stimuli.

Images and captions were taken from the training and validation sets of the COCO dataset (**Lin et al., 2014**). This dataset contains five matching captions for each image, of which we only considered the shortest one in order to fit on the screen and to ensure a comparable length for all captions. Spelling errors were corrected manually. As our training set, a random subset of images and another random subset of captions were selected for each subject. All these stimuli were presented only a

single time (except for the case of one-back target trials, for which the second presentation was, however, excluded during the preprocessing of the data). Information on the number of training stimuli for each subject is shown in *Figure 1* Panel B. Additionally, a shared subset of 140 stimuli (70 images and 70 captions) was presented repeatedly to each subject in order to reduce noise, serving as our test set (on average: 26 times, min: 22, max: 31) (contrary to the training stimuli which were randomly selected from the COCO dataset, the 70 test stimuli were chosen by hand to avoid, including multiple scenes that could match the same semantic description. The 70 chosen images, as well as their 70 corresponding captions, constituted the test set). These stimuli were inserted randomly between the training stimuli.

Note that for each stimulus presented to the subject (e.g. an image), we also have access to the corresponding stimulus in the other modality (the corresponding caption from the COCO dataset), allowing us to estimate model features based on both modalities (vision model features extracted from the image and language model features extracted from the corresponding caption) as well as multimodal features extracted from both the image and the caption.

In addition to these perceptual trials, there were three imagery trials for each subject (see also *Figure 1* Panel C). Prior to the first fMRI scanning session, each subject was presented with a set of 20 captions (manually selected to be diverse and easy to visualize) that were not part of the perceptual trials, and they selected three captions for which they felt comfortable imagining a corresponding image. Then, they learned a mapping of each caption to a number (1, 2, and 3) so that they could be instructed to perform mental imagery of a specific stimulus, without having to present them with the caption again. The imagery trials occurred every second run, either at the beginning or the end of the run, so that each of the three imagery conditions were repeated on average 26 times (min: 23, max: 29). At the start of the imagery trial, the imagery instruction number was presented for 2 s, then there was a 1 s fixation period followed by the actual imagery period during which a light gray box was depicted for 10 s on a dark gray background (the same background that was also used for trials with perceptual input). The light gray box was meant to represent the area in which the mental image should be 'projected.' At the end of the experiment, the subjects drew sketches of the images they had been imagining during the imagery trials.

We report dataset quality metrics on head motion and intersession alignment in Appendix 1.

## fMRI preprocessing

Preprocessing of the fMRI data was performed using SPM12 (*Ashburner et al., 2014*) via nipype (*Gorgolewski et al., 2011*). We applied slice time correction and realignment for each subject. Each session was coregistered with an anatomical scan of the respective subject's first session (downsampled to 2 mm$^3$). We created and applied explicit gray matter masks for each subject based on their anatomical scans using a maximally lenient threshold (probability >0).

In order to obtain beta values for each stimulus, for each subject, we fit a GLM (using SPM12) on data from all sessions. We included regressors for train images, train captions, test images, test captions, imagery trials, fixations, blank screens, button presses, and one-back target trials. One-back target trials, as well as trials in which the participant pressed the button were excluded in the calculation of all training and test stimulus betas. As output of these GLMs, we obtained beta values for each training and test caption and image, as well as the imagery trials.

Finally, we transformed the volume-space data to surface space Freesurfer (*Fischl, 2012*). We used trilinear interpolation and the fsaverage template in the highest possible resolution (163,842 vertices on each hemisphere) as target.

## Modality-agnostic decoders

The multimodal nature of our dataset allowed for the training of modality-agnostic decoders. We trained decoders by fitting ridge regression models that take fMRI beta values as input and predict latent representations extracted from a pretrained deep learning model. Further details on decoder training can be found in *Figure 2* as well as Appendix 2.

While modality-specific decoders are trained only on brain imaging data of a single modality, modality-agnostic decoders are trained on brain imaging data from multiple modalities and, therefore, allow for decoding of stimuli irrespective of their modality.

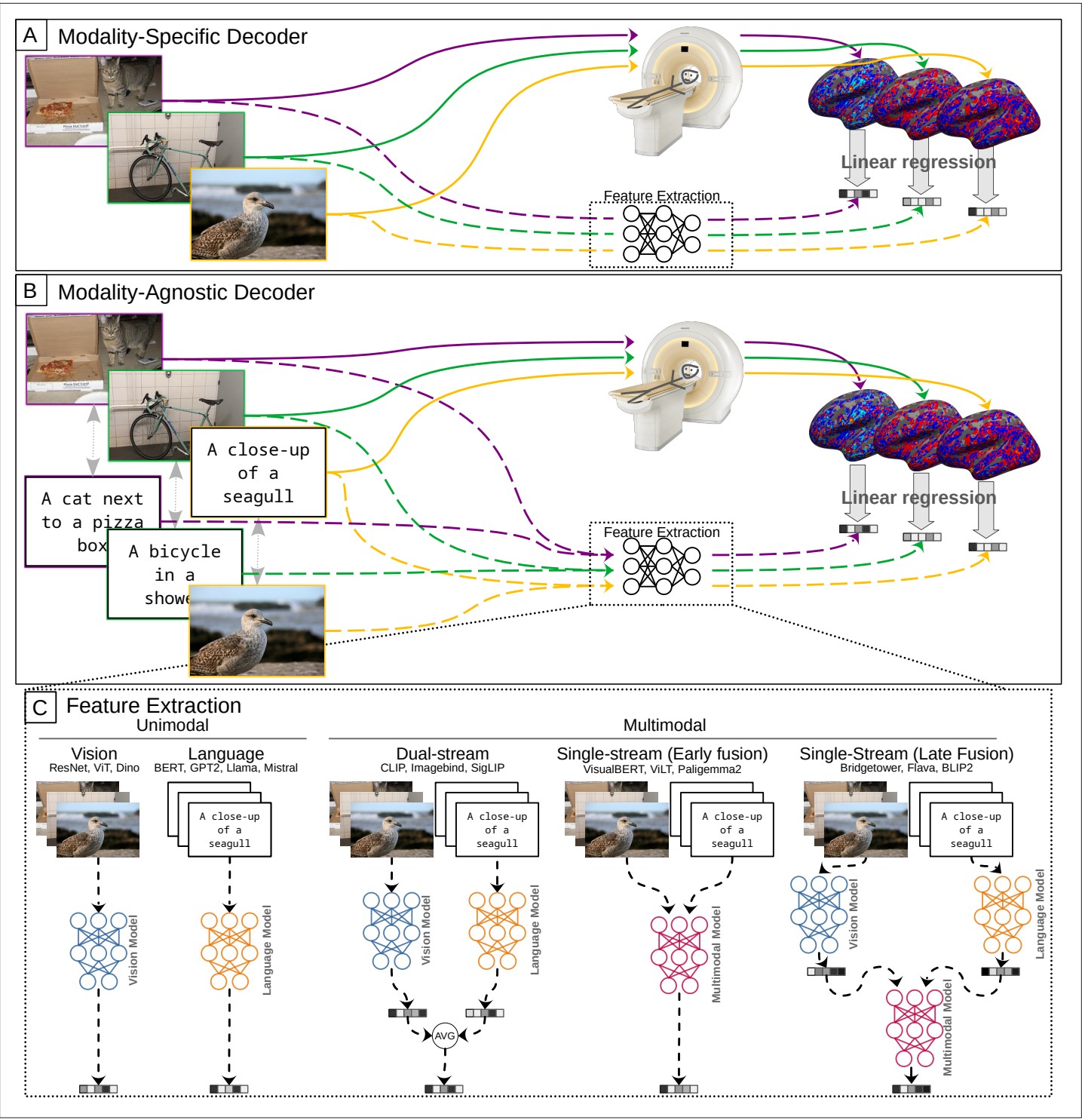

**Figure 2.** Training of modality-specific and modality-agnostic decoders. (**A**) Modality-specific decoders are trained on fMRI data of one modality (e.g. subjects viewing images) by mapping it to features extracted from the same stimuli. (**B**) Modality-agnostic decoders are trained jointly on fMRI data of both modalities (subjects viewing images and captions). (**C**) To train decoders, features can be either extracted unimodally from the corresponding images or captions, or by creating multimodal features based on both modalities. For example, to train a modality-agnostic decoder based on features from a unimodal language model, we map the fMRI data of subjects viewing captions to features extracted from the respective captions using this language model, as well as the fMRI data of subjects viewing images to features extracted by the language model from the corresponding captions. We can also train modality-specific decoders on features from another modality, for example, by mapping fMRI data of subjects viewing images to features extracted from the corresponding captions using a language model.

More specifically, in our case, the modality-specific decoders are trained on fMRI beta values from one stimulus modality, e.g., when subjects were watching images (*Figure 2* panel A). Conversely, modality-agnostic decoders are trained jointly using fMRI data from both stimulus modalities (images and captions; *Figure 2* panel B). For all decoders, the features that serve as regression targets can either be unimodal (e.g. extracted from images using a vision model) or multimodal (e.g. extracted from both stimulus modalities using a multimodal model, *Figure 2* panel C).

We considered features extracted from a range of vision, language, and multimodal models: For vision features, we considered ResNet (*He et al., 2016*), ViT (*Dosovitskiy et al., 2020*), and DINOv2 *Oquab et al., 2023*; for language features BERT (*Devlin et al., 2019*), GPT2 (*Radford et al., 2019*), Llama2 (*Touvron et al., 2023*), mistral and mixtral (*Jiang et al., 2023*). Regarding multimodal features, we extracted features from VisualBERT (*Li et al., 2019*), BridgeTower (*Xu et al., 2023*), ViLT (*Kim et al., 2021*), CLIP (*Radford et al., 2021*), ImageBind (*Girdhar et al., 2023*), Flava (*Singh et al., 2022*), Blip2 (*Li et al., 2023*), SigLip (*Zhai et al., 2023*), and Paligemma2 (*Steiner et al., 2024*). In order to estimate the effect of model training, we further extracted features from a randomly initialized Image-Bind model as a baseline. Further details on feature extraction can be found in Appendix 2.

All decoders were evaluated on the held-out test data (140 stimuli, 70 captions, and 70 images) using pairwise accuracy calculated using cosine distance. Prior to calculating the pairwise accuracy, the model predictions for all stimuli were standardized to have a mean of 0 and a standard deviation of 1 (In the case of imagery decoding, the model predictions were standardized separately). In the case of cross-modal decoding (e.g. mapping an image stimulus into the latent space of a language model), a trial was counted as correct if the caption corresponding to the image (according to the ground truth in COCO) was closest.

*Figure 3* provides an overview of the evaluation metrics. A modality-specific decoder for images can be evaluated on its ability to decode images (Panel A, top) and in a cross-decoding setup for captions (Panel B, bottom). In the same way, we can compute the respective evaluation metrics for modality-specific decoders trained on captions. For the case of modality-agnostic decoders, we evaluate performance for decoding both images and captions using the same single decoder that is trained on both modalities (Panel C).

## Results

### Modality-agnostic decoders

We first compared the performance of modality-specific and modality-agnostic decoders that are trained on the whole brain fMRI data based on different unimodal and multimodal features. The average pairwise accuracy scores are presented in *Figure 4*. *Figure 5* presents pairwise accuracy scores separately for decoding images and for decoding captions. Results for individual subjects can be found in Appendix 6.

When analyzing the average decoding accuracy (*Figure 4*), we find that modality-agnostic decoders perform better than modality-specific decoders. To support this observation, we performed two repeated measures ANOVAs (grouping the data by subject), once comparing the average accuracy values of modality-agnostic decoders (mean value: 81.86%) with those of modality-specific decoders trained on images (mean value: 78.15%), and once comparing modality-agnostic decoders to modality-specific decoders trained on captions (mean value: 72.52%). In both cases, the accuracy values for the two decoder types were significantly different. When comparing modality-agnostic decoders to modality-specific decoders trained on images: decoder_type : $\beta = 0.037, SE = 0.004, p < 1 \cdot 10^{-18}$; and when comparing to modality-specific decoders trained on captions: decoder_type : $\beta = 0.093, SE = 0.005, p < 1 \cdot 10^{-93}$. This high performance (which can be attributed to the large training dataset used by modality-agnostic decoders) is achieved despite the additional challenge of not knowing the modality of the stimulus the subject was seeing.

Furthermore, we observed that modality-agnostic decoders based on the best multimodal features (imagebind: $85.71\% \pm 2.58\%$) do not perform substantially better than decoders based on the best language features (GPT2-large: $85.31\% \pm 2.83\%$) and only slightly better than decoders trained on the best vision features (Dino-giant: $82.02\% \pm 2.43\%$). This result suggests that high-performing modality-agnostic decoders do not necessarily need to rely on multimodal features; features extracted from language models can lead to equally high performance. When comparing

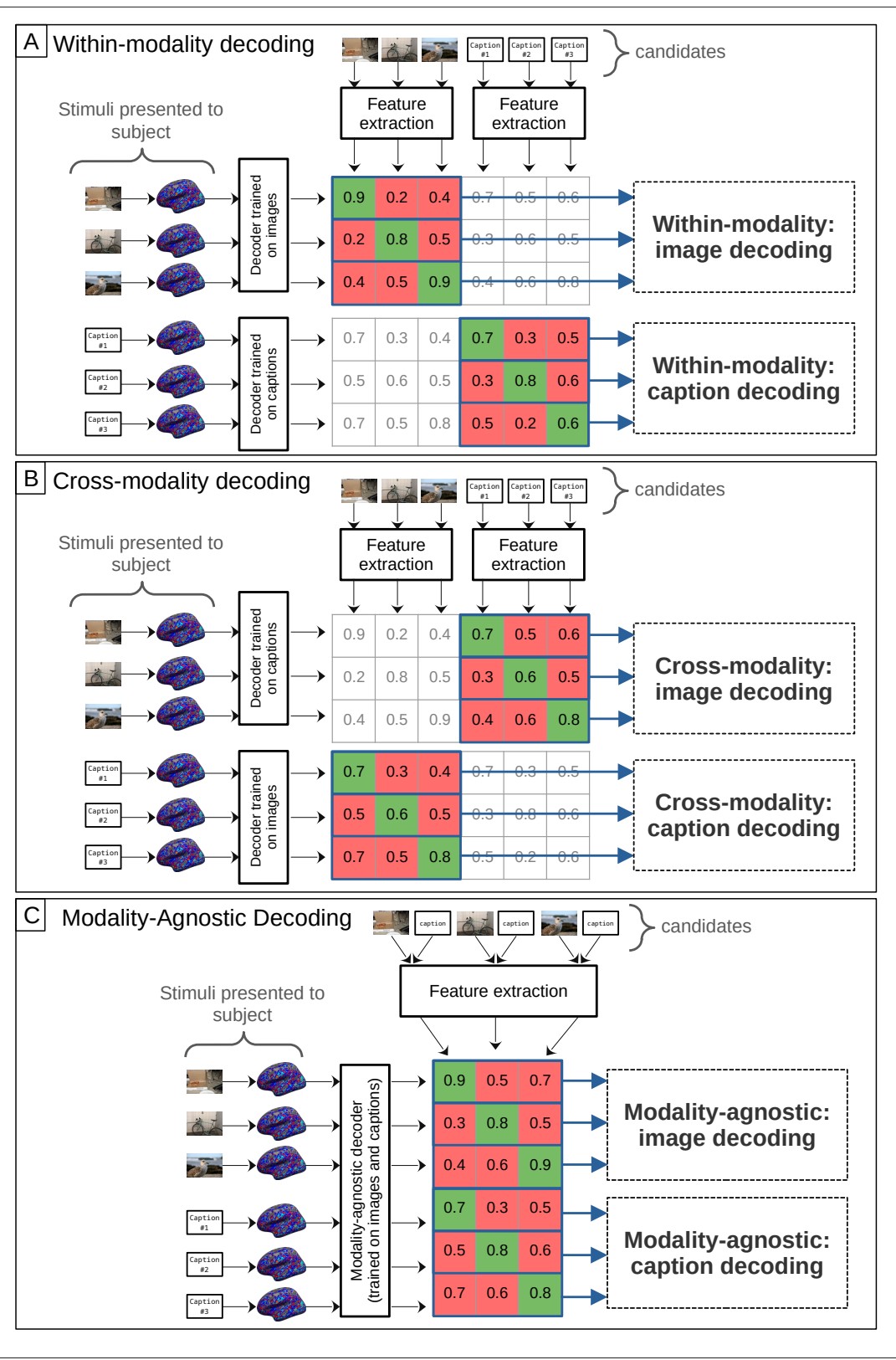

**Figure 3.** Evaluation of modality-specific and modality-agnostic decoders. The matrices display cosine similarity scores between features extracted from the candidate stimuli and features predicted by the decoder. The evaluation metric is pairwise accuracy, which is calculated row-wise: For a given matrix row, we compare the similarity score of the target stimulus on the diagonal (in green) with the similarity scores of all other candidate

*Figure 3 continued on next page*

*Figure 3 continued*

stimuli (in red). (**A**) Within-modality decoding metrics of modality-specific decoders. To compute within-modality accuracy for image decoding, a modality-specific decoder trained on images is evaluated on all stimuli that were presented as images. To compute within-modality accuracy for caption decoding, a modality-specific decoder trained on captions is evaluated on all caption stimuli. (**B**) Cross-modality decoding metrics of modality-specific decoders. To compute cross-modality accuracy for image decoding, a modality-specific decoder trained on captions is evaluated on all stimuli that were presented as images. To compute cross-modality accuracy for caption decoding, a modality-specific decoder trained on images is evaluated on all caption stimuli. (**C**) Metrics for modality-agnostic decoders. To compute modality-agnostic accuracy for image decoding, a modality-agnostic decoder is evaluated on all stimuli that were presented as images. The same decoder is evaluated on caption stimuli to compute modality-agnostic accuracy for caption decoding. Here, we show feature extraction based on unimodal features for modality-specific decoders and based on multimodal features for the modality-agnostic decoder. In practice, the feature extraction can be unimodal or multimodal for any decoder type (see also *Figure 2*).

the different architecture types of models for multimodal feature extraction (dual stream vs. single stream with early fusion vs. single stream with late fusion; cf. Panel C in *Figure 2*), we only observed a slight performance disadvantage for single-stream models with early fusion (Mean accuracy values for dual stream models: 85.04%; for single stream models with early fusion: 81.01%; for single stream models with late fusion: 83.63%). We confirmed this observation with a repeated measures ANOVA (grouping the data by subject), comparing the decoding accuracy values of modality-agnostic decoders based on different families of multimodal features. The only significant effect was: model_family_single_stream_early_fusion : $\beta = -0.04, SE = 0.011, p < 1e - 3$.

When analyzing the performance specifically for decoding images (*Figure 5*, top), we find that modality-agnostic decoders perform as well as the modality-specific decoders trained on images (green bars in the top row are at the same level as the orange bars in *Figure 5*). We found no statistically significant difference in their performances. A repeated measures ANOVA (grouping the data by subject), comparing the image decoding accuracy values of modality-agnostic decoders with those of modality-specific decoders trained on images resulted in a p-value of $p = 0.73$ for the effect of the decoder_type.

Furthermore, modality-agnostic decoders even outperform modality-specific decoders trained on captions for decoding captions (purple bars in the bottom row are lower than green bars in *Figure 5*). This was supported by a repeated measures ANOVA (grouping the data by subject), comparing the caption decoding accuracy values of modality-agnostic decoders with those of modality-specific decoders trained on captions. The result was: decoder_type : $\beta = 0.036, SE = 0.006, p < 1 \cdot 10^{-8}$. In other words, even if we know that a brain pattern was recorded in response to the subject reading a caption,

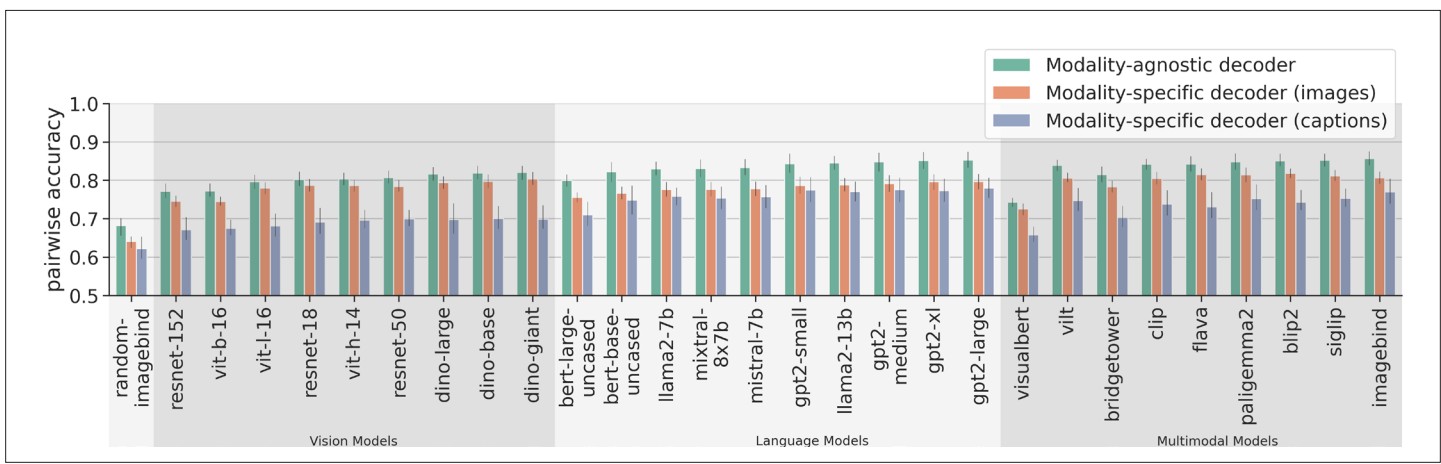

**Figure 4.** Average decoding scores for modality-agnostic decoders (green), compared to modality-specific decoders trained on data from subjects viewing images (orange) or on data from subjects viewing captions (purple). The metric is pairwise accuracy (see also *Figure 3*). Error bars indicate 95% confidence intervals calculated using bootstrapping. Chance performance is at 0.5.

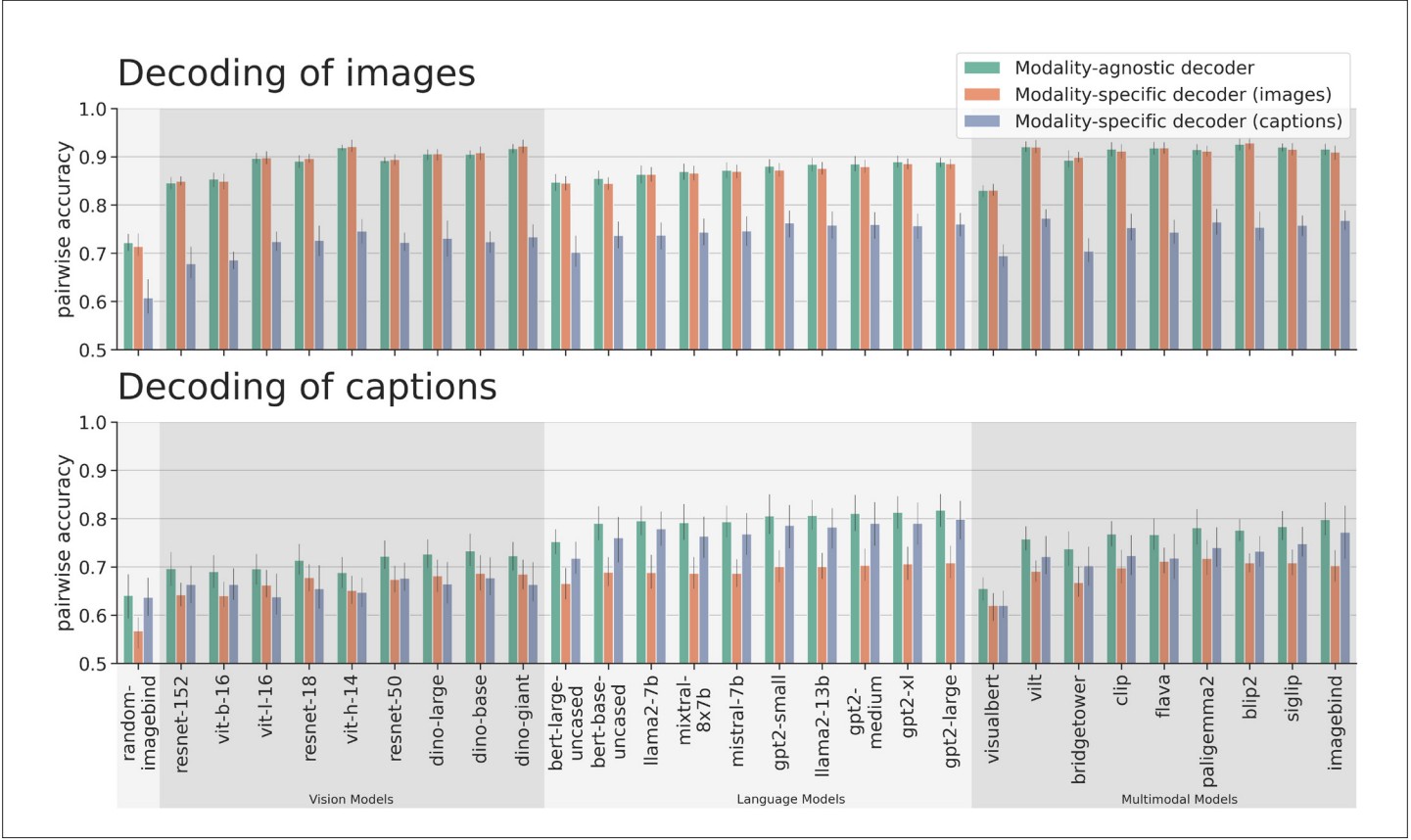

**Figure 5.** Decoding accuracy for decoding images (top) and for decoding captions (bottom). The orange bars in the top row indicate within-modality decoding scores for images, the purple bars in the bottom row indicate within-modality decoding scores for captions. The purple bars in the top row indicate cross-decoding scores for images, the orange bars in the bottom row indicate cross-decoding scores for captions (see also *Figure 3*). Error bars indicate 95% confidence intervals calculated using bootstrapping. Chance performance is at 0.5.

we are more likely to decode it accurately if we choose to apply a decoder trained using both modalities than if we apply the appropriate decoder, trained only on captions.

Finally, we found that the cross-modal decoding performance for decoding visual stimuli (images) using decoders trained on captions (mean value: 73.31%) is higher than the cross-modal decoding performance in the other direction (mean value: 68.03%), corroborating similar results from *Tang et al., 2023a* on movies and audio books (purple bars in top row are higher than orange bars in bottom row in *Figure 5*). We performed a repeated measures ANOVA (grouping the data by subject) to compare the pairwise accuracy values of both cross-decoding directions and found that the difference was significant: $\text{decoder\_type} : \beta = 0.053, SE = 0.004, p < 1 \cdot 10^{-31}$ (the decoder_type variable is indicating whether the decoder is trained on images or captions).

## Qualitative decoding results

To obtain a better understanding of the decoding performance of the modality-agnostic decoders, we inspected the decoding results for five randomly selected test stimuli. We created a large candidate set of 41,118 stimuli by combining the test stimuli and the training stimuli from all subjects. For each stimulus, we ranked these candidate set stimuli based on their similarity to the predicted feature vector. As the test stimuli were shared among all subjects, we could average the prediction feature vectors across subjects to obtain the best decoding results.

*Figure 6* presents the results for decoding images using a modality-agnostic decoder trained on ImageBind features. We display the target stimulus along with the top-5 ranked candidate stimuli. We can observe some clear success cases (the train in the first row, a person eating pizza in the last row) but also failure cases (the teddy bear decoded as pizza). For the other stimuli, some aspects, such as the high-level semantic class (e.g. vehicle, animal, sports) are correctly decoded: For the cars on the

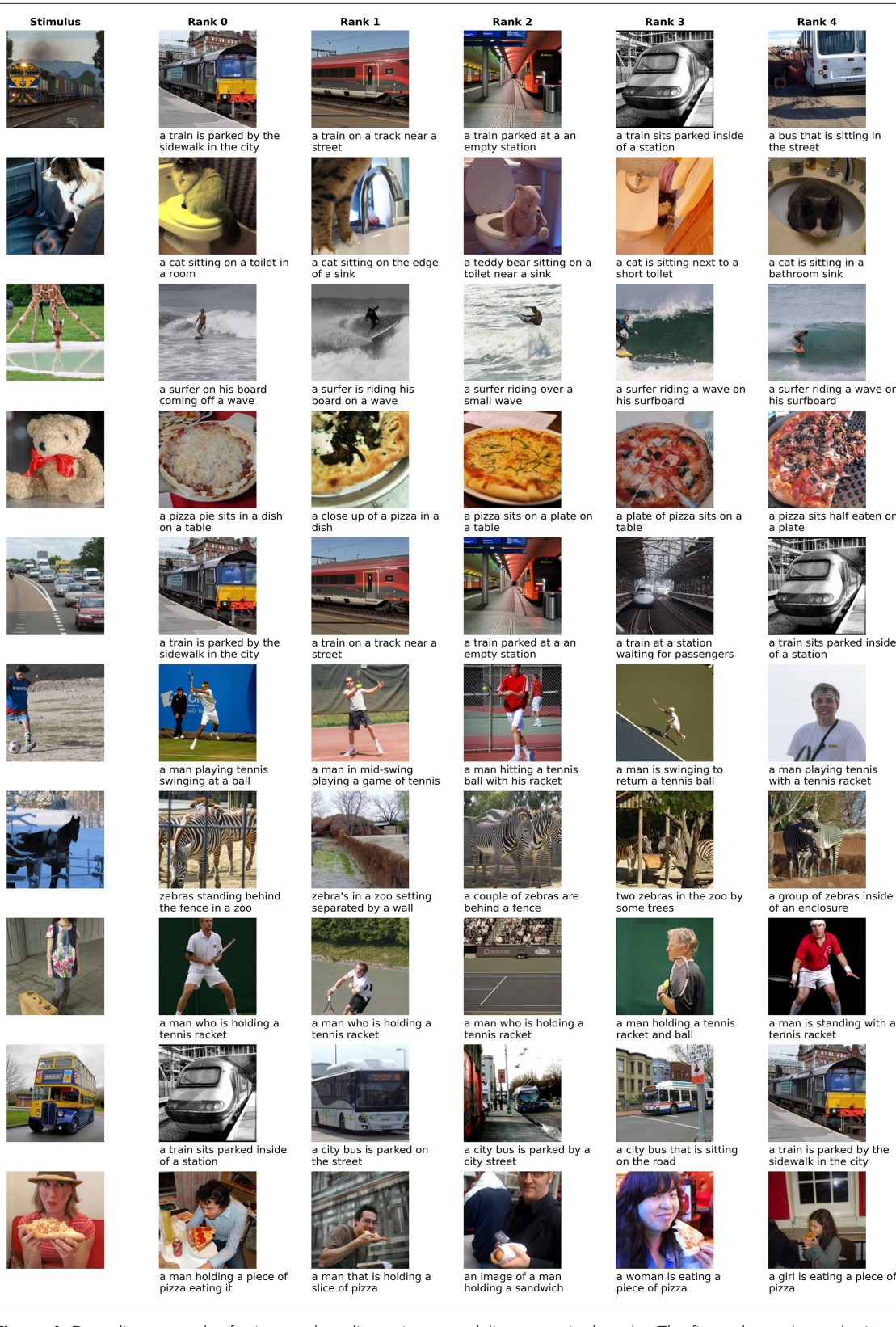

**Figure 6.** Decoding examples for image decoding using a modality-agnostic decoder. The first column shows the image the subject was seeing and the five following columns show the candidate stimuli with highest similarity to the predicted features, in descending order. We display both the image and the caption of the candidate stimuli because the decoder is based on multimodal features that are extracted from both modalities. All images were taken from the CoCo dataset (*Lin et al., 2014*).

highway (fifth row), the top-ranked images depict trains, which are also vehicles. For the dog (second row), the top images contain cats of similar colors. The footballer in the sixth row is decoded as a tennis player, and the horses of the seventh row are decoded as zebras. Regarding the giraffe (third row), the model appears to have picked up on the fact that there was a body of water depicted in the image.

Note that these qualitative results are not directly comparable with previous work on retrieval or reconstruction using the NSD dataset (*Allen et al., 2022*; *Lin et al., 2022*; *Takagi and Nishimoto, 2023*; *Ozcelik and VanRullen, 2023*), as our data was collected on a 3T MRI scanner with lower signal-to-noise ratio than NSD's 7T MRI scanner.

The ranking results for decoding captions are depicted in *Figure 7*. The results are somewhat similar to the image decoding results. Stimuli that were decoded successfully when presented as image, such as the train are also decoded successfully when presented as caption; cases that were failures in the case of image decoding (e.g. the teddy bear) also fail here. However, the top-ranked stimuli for the dog (second row) do not always contain animals (the decoder seems to have picked up on the presence of a vehicle in the caption), and the horses of the seventh row do not get decoded correctly. The cars on the highway (fifth row) get decoded more accurately than during image decoding.

We additionally provide qualitative results for modality-specific decoders in Appendix 3. These results generally reflect the observations from the quantitative results: Modality-agnostic decoders perform similarly to modality-specific decoders evaluated in a within-modality decoding setup, but substantially better than modality-specific decoders when evaluated in cross-decoding setups.

## Modality-invariant regions

Modality-agnostic decoders perform best when they can leverage modality-invariant representations. To provide insight into the spatial organization of such modality-invariant representations in the brain, we performed a surface-based searchlight analysis.

Modality-invariant regions should contain patterns that generalize between stimulus modalities. Therefore, such regions should allow for decoding of stimuli in both modalities using a decoder that is trained to pick up on modality-invariant features, i.e., the decoding performance for images and captions of a modality-agnostic decoder should both be above chance. However, as a modality-agnostic decoder is trained on stimuli from both modalities, it could have learned to leverage certain features to project stimuli from one modality and different features to project stimuli from the other modality. We added two conditions to control that the representations directly transfer between the modalities by additionally training two modality-specific decoders and evaluating them according to their cross-decoding performance, i.e., we require that their decoding performance in the modality they were not trained on is above chance. These four conditions are summarized at the top of *Figure 8*.

We used ImageBind features for these searchlight analyses as they led to the highest decoding performance when using the whole brain data. The decoders were trained based on the surface projection of the fMRI beta values. For each vertex, we defined a searchlight with a fixed size by selecting the 750 closest vertices, corresponding to an average radius of ~9.4 mm (details on how this size was selected are outlined in Appendix 5).

We trained and evaluated a modality-agnostic decoder and modality-specific decoders for both modalities on the beta values for each searchlight location and each subject, providing us with decoding accuracy scores for each location on the cortex. Then we performed one-tailed t-tests to identify locations in which the decoding performance is above chance ($acc > 0.5$). We aggregated all four comparisons by taking the minimum of the four t-values at each spatial location. Finally, we performed threshold-free cluster enhancement ([, TFCE,)]smith_threshold-free_2009 to identify modality-invariant ROIs (*Figure 8*, bottom) (we used the default hyperparameters of $h$=2 and e=1 for surface-based TFCE *Jenkinson et al., 2012*).

To estimate the statistical significance of the resulting clusters, we performed a permutation test, combined with a bootstrapping procedure to estimate a group-level null distribution (see also *Stelzer et al., 2013*). For each subject, we evaluated the decoders 100 times with shuffled labels to create per-subject chance-level results. Then, we randomly selected one of the 100 chance-level results for each of the six subjects and calculated group-level statistics (TFCE values) the exact same way as described in the preceding paragraph. We repeated this procedure 10,000 times resulting in 10,000 permuted group-level results. We ensured that every permutation was unique, i.e., no two permutations were

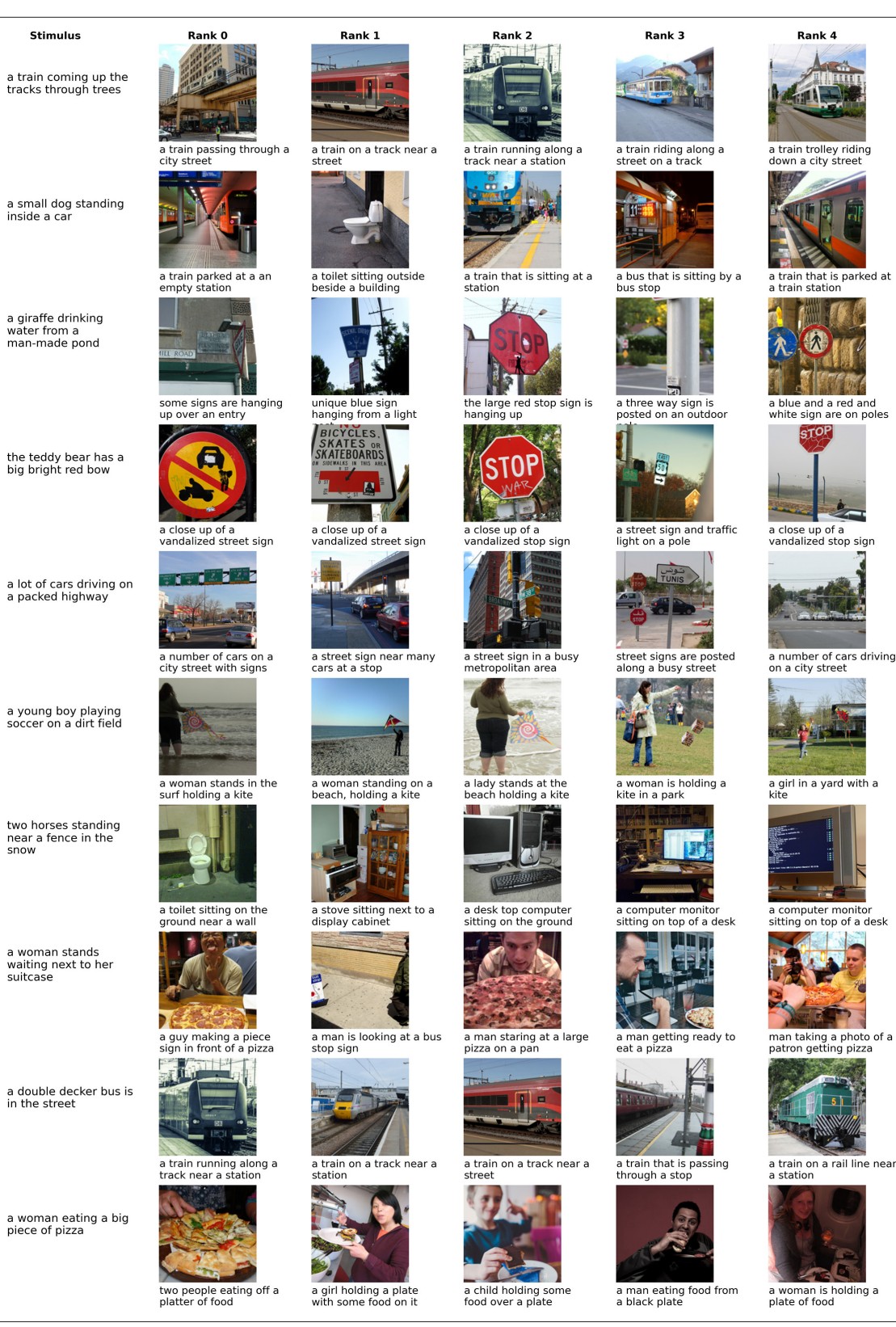

**Figure 7.** Decoding examples for caption decoding using a modality-agnostic decoder. For details, see caption of *Figure 6*. All images were taken from the CoCo dataset (*Lin et al., 2014*).

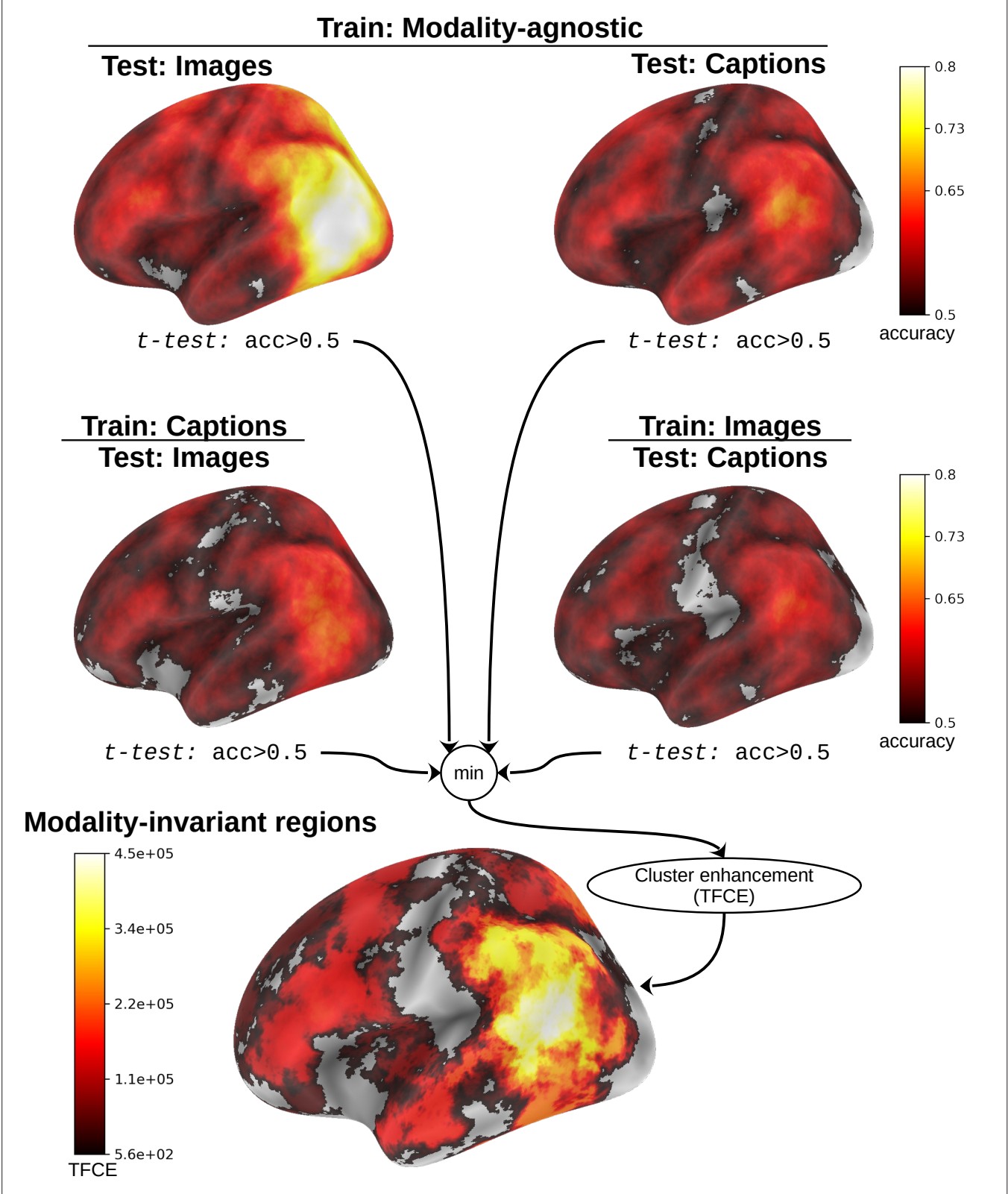

**Figure 8.** Searchlight method to identify modality-invariant ROIs. The top plots show performance (pairwise accuracy averaged over subjects) of modality-agnostic decoders for decoding images (top left) and decoding captions (top right). In the second row, we display cross-decoding performances: On the left, modality-specific decoders trained on captions are evaluated on images. On the right, modality-specific decoders trained on images are evaluated on captions. We identified modality-invariant ROIs as clusters in which all 4 decoding accuracies are above chance by taking

*Figure 8 continued on next page*

*Figure 8 continued*

the minimum of the respective t-values at each location, then performed threshold-free cluster enhancement (TFCE) to calculate cluster values. The plot only shows left medial views of the brain to illustrate the method; different views of all resulting clusters are shown in *Figure 9*.

based on the same combination of selected chance-level results. Based on this null distribution, we calculated p-values for each vertex by calculating the proportion of sampled permutations where the TFCE value was greater than the observed TFCE value. To control for multiple comparisons across space, we always considered the maximum TFCE score across vertices for each group-level permutation (*Smith and Nichols, 2009*).

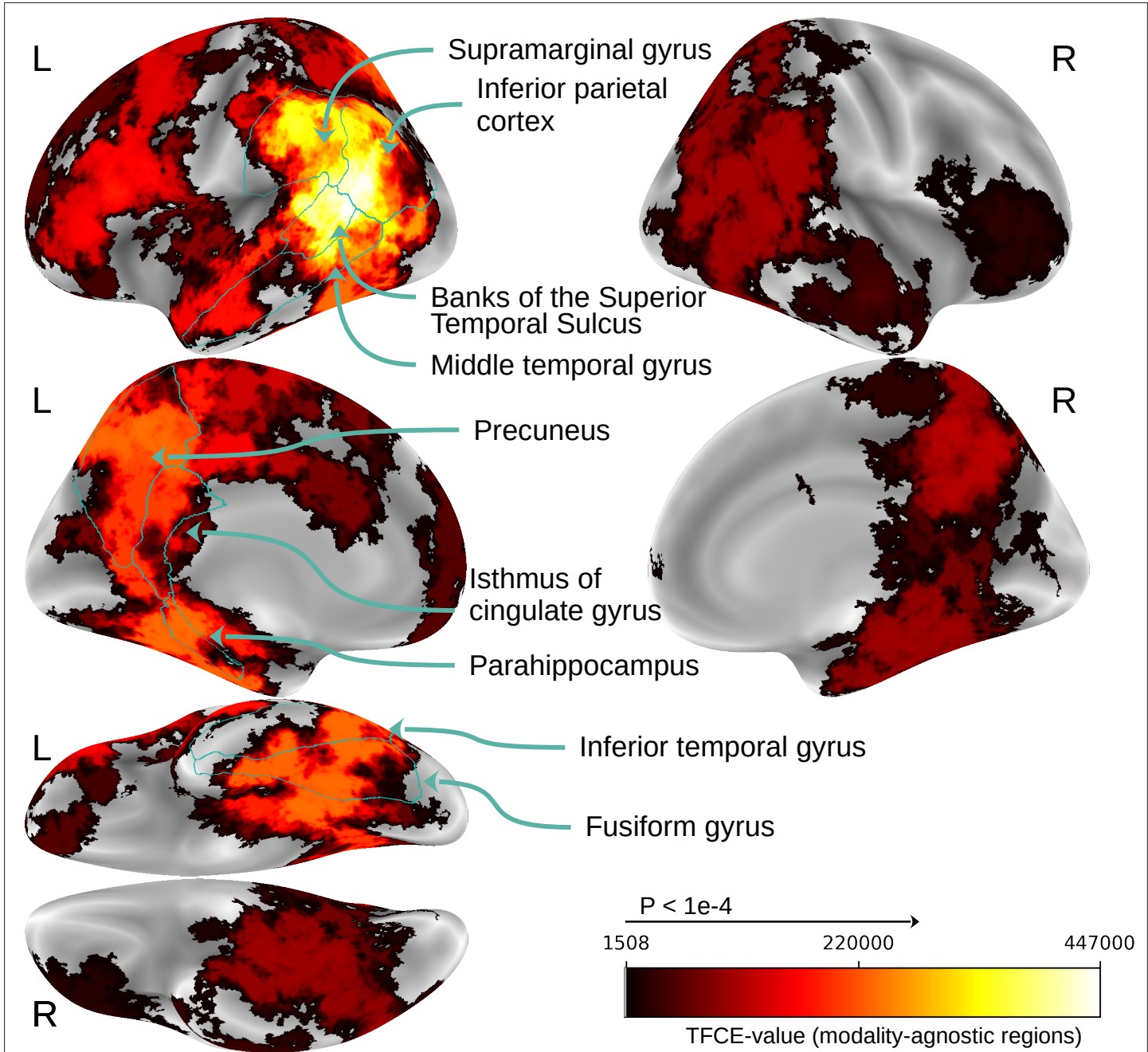

**Figure 9.** Searchlight results for modality-invariant regions. Maps thresholded at threshold-free cluster enhancement (TFCE) value of 1508, which is the significance threshold value for which $p < 10^{-4}$ based on the permutation testing. Regions with the highest cluster values are outlined and annotated based on the Desikan-Killiany atlas (*Desikan et al., 2006*).

The results of the surface-based searchlight analysis are presented in *Figure 9*. The analysis revealed that modality-invariant patterns are actually widespread across the brain, especially on the left hemisphere. Peak cluster values were found in the left supramarginal gyrus, inferior parietal gyrus and posterior superior temporal sulcus. Regions belonging to the precuneus, isthmus of the cingulate gyrus, parahippocampus, middle temporal gyrus, inferior temporal gyrus, and fusiform gyrus also showed high cluster values.

## Imagery decoding

Finally, we evaluated the ability of decoders trained on the fMRI data with perceptual input to decode stimuli during the imagery conditions.

For a modality-agnostic decoder trained on the whole-brain data, the imagery pairwise decoding accuracy reaches 84.48% (averaged across subjects and model features) when using the three imagery stimuli as the candidate set (note that we used the ground-truth caption and corresponding image from COCO in this candidate set, and not the sketches drawn by the subjects). When the whole test set is added to the candidate set (in total: 73 stimuli), the average pairwise accuracy drops to 72.47%. This substantial drop in performance is most likely explained by the fact that the predicted features for the imagery trials were standardized using only three stimuli, and this transformation emphasized differences that enabled distinguishing the three imagery trials but do not generalize to the larger test set (we also attempted decoding without standardization of the predicted feature vectors, but this led to much lower performance).

As expected, we also found that modality-agnostic decoders are better suited for imagery decoding than modality-specific decoders. We compared the imagery decoding accuracy of both decoder types, taking into account the results for all features and all subjects. To this end, we performed two repeated measures ANOVAs (grouping the data by subject), once comparing the accuracy values of modality-agnostic decoders with those of modality-specific decoders trained on images, and once comparing modality-agnostic decoders to modality-specific decoders trained on captions. The average decoding accuracies were 69.42% for a modality-specific decoder trained on images and 70.02% for a modality-specific decoder trained on captions (vs. 72.47% for a modality-agnostic decoder, as mentioned above). In both comparisons, the accuracy values for the two decoder types were significantly different (when comparing modality-agnostic decoders to modality-specific decoders trained on images: $decoder\_type : \beta = 0.03, SE = 0.011, p < 0.01$; and when comparing to modality-specific decoders trained on captions: $decoder\_type : \beta = 0.024, SE = 0.012, p < 0.04$).

Appendix 7 presents qualitative decoding results for the imagery trials for each subject as well as the sketches of the mental images drawn at the end of the experiment. As expected, the results are worse than those for perceived stimuli, but for several subjects, it was possible to decode some major semantic concepts.

We further computed the imagery decoding accuracy during the searchlight analysis. *Figure 10* shows the result clusters for decoding imagery (using the whole test set + the three imagery trials as potential candidates). To assess statistical significance, we employed the same permutation testing procedure with bootstrapping as used for identifying modality-invariant regions (10,000 permutations based on random combinations of 100 chance distributions for each subject).

We observe that many regions that were found to contain modality-invariant patterns (*Figure 9*) are also regions in which decoding of mental imagery is possible.

One main difference is that the imagery decoding clusters appear to be less left-lateralized than the modality-invariant region clusters (peak cluster values can be found both on the right inferior parietal cortex and bilaterally in the precuneus). To estimate the overlap of the regions allowing for imagery decoding and modality-invariant regions, we calculated the correlation between the TFCE values that were used for identifying modality-invariant regions (*Figure 9*) and the TFCE values for imagery decoding (*Figure 10*). The Pearson correlation score for the left hemisphere is 0.41 ($p < 1e - 8$), and for the right hemisphere, 0.62 ($p < 1e - 8$). Importantly, these correlation scores are substantially higher when compared to the correlation with decoding accuracy of modality-specific decoders: The correlation between the TFCE values for imagery decoding and TFCE values for image decoding of a modality-specific decoder trained on images is 0.28 on the left hemisphere and 0.40 on the right hemisphere. When using TFCE values based on the caption decoding accuracy of a modality-specific decoder trained on captions, we obtain 0.19 on the left hemisphere and 0.45 on the right hemisphere.

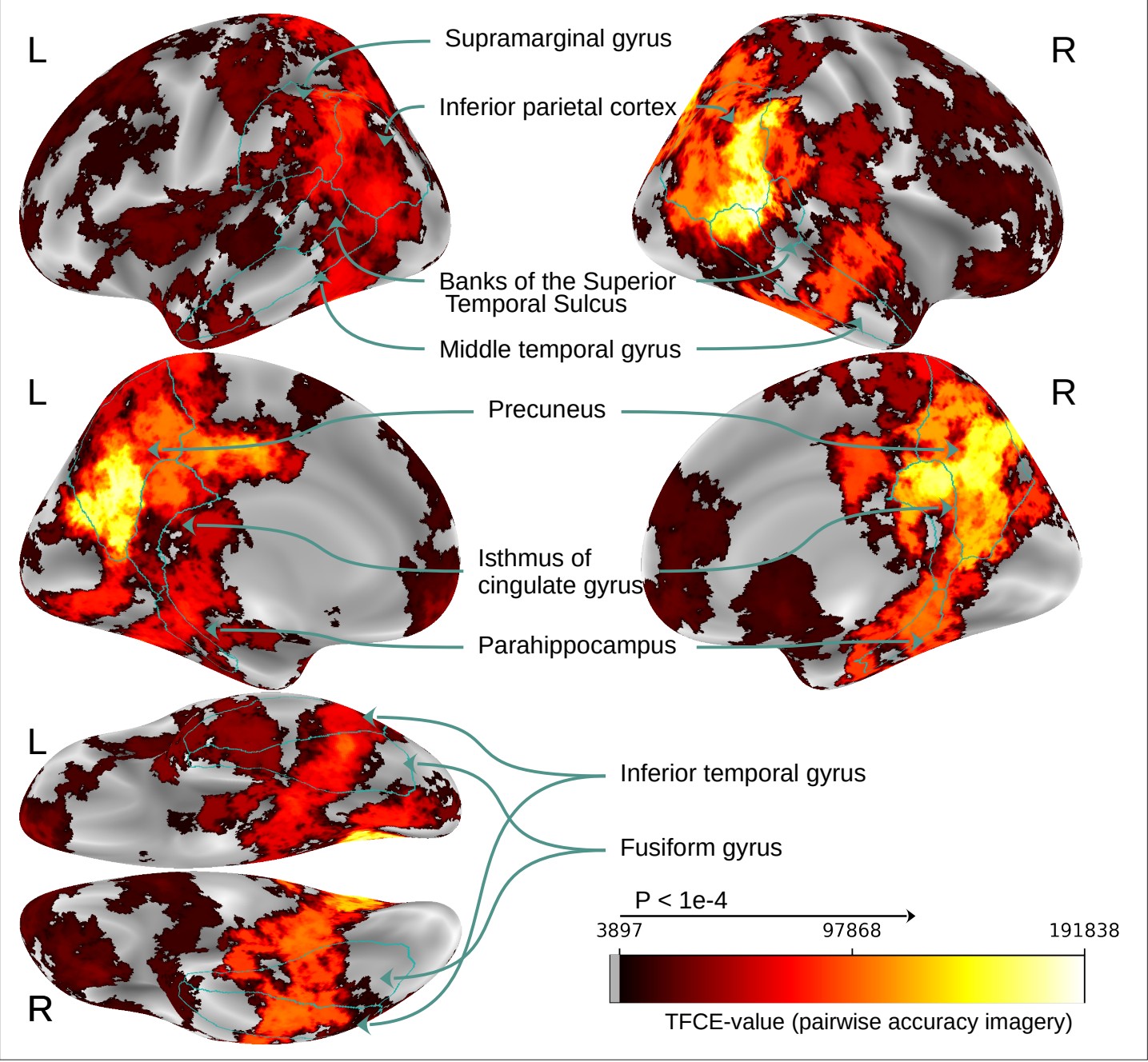

**Figure 10.** Searchlight results for imagery decoding. Maps thresholded at threshold-free cluster enhancement (TFCE) values that surpass the significance threshold of $p < 10^{-4}$ based on a permutation test. Maps thresholded at TFCE value of 3897, which is the significance threshold value for which $p < 10^{-4}$ based on the permutation testing. We used the pairwise accuracy for imagery decoding using the large candidate set of 73 stimuli. We outlined the same regions as in *Figure 9* to facilitate comparison.

## Discussion

In this work, we introduced modality-agnostic decoders, which are trained to decode stimuli from different modalities using a single model. Such modality-agnostic decoders are specifically trained to pick up on modality-invariant representations, enabling a performance increase over modality-specific decoders when decoding linguistic stimuli in the form of captions as well as when decoding mental imagery. Previous work on decoding with large-scale brain imaging datasets has largely focused on modality-specific cortices, such as the visual cortex for decoding natural scenes (*Shen et al., 2019;*

*Beliy et al., 2019*; *Lin et al., 2022*; *Takagi and Nishimoto, 2023*; *Ozcelik and VanRullen, 2023*). Our work demonstrates the advantages of leveraging higher-level representations that are shared between modalities, and which we found to be present in widespread areas of the human cortex. In the following, we will further discuss the nature and location of the modality-invariant representations in light of previous experimental and theoretical work.

The results of our searchlight analysis suggest that modality-invariant representations are found in temporal, parietal, and frontal regions. Peak cluster values are found on the border of the temporal and parietal cortices on the left hemisphere (*Figure 9*). All areas with high cluster values confirm findings from previous studies: The left precuneus (*Shinkareva et al., 2011*; *Fairhall and Caramazza, 2013*; *Popham et al., 2021*; *Handjaras et al., 2016*), posterior cingulate/retrosplenial cortex (*Fairhall and Caramazza, 2013*; *Handjaras et al., 2016*), supramarginal gyrus (*Shinkareva et al., 2011*), inferior parietal cortex (*Man et al., 2012*; *Vandenberghe et al., 1996*; *Shinkareva et al., 2011*; *Devereux et al., 2013*; *Popham et al., 2021*; *Simanova et al., 2014*; *Handjaras et al., 2016*), superior temporal sulcus (*Man et al., 2012*), middle temporal gyrus (*Vandenberghe et al., 1996*; *Shinkareva et al., 2011*; *Fairhall and Caramazza, 2013*; *Devereux et al., 2013*; *Handjaras et al., 2016*), inferior temporal gyrus (*Vandenberghe et al., 1996*; *Shinkareva et al., 2011*; *Fairhall and Caramazza, 2013*; *Simanova et al., 2014*; *Handjaras et al., 2016*), fusiform gyrus (*Vandenberghe et al., 1996*; *Moore and Price, 1999*; *Bright et al., 2004*; *Shinkareva et al., 2011*; *Fairhall and Caramazza, 2013*; *Simanova et al., 2014*), and parahippocampus (*Vandenberghe et al., 1996*). However, previous studies have led to contradicting results regarding the locality of modality-invariant regions (they were identifying varying subsets of these regions; see also Appendix 4), probably due to the limited number and artificial nature of stimuli employed. Our method identified *almost all* of the previously proposed regions as regions with modality-invariant patterns, highlighting the advantage of our searchlight decoding approach and the large multimodal dataset in which subjects are viewing photographs of complex natural scenes and reading full English sentences. (The left superior occipital gyrus was not identified in our study, but in previous studies by *Shinkareva et al., 2011*; *Vandenberghe et al., 1996*. However, we found that a major part of the left superior occipital sulcus represents modality-invariant information. Furthermore, *Jung et al., 2018* found modality-invariant patterns in the right superior frontal gyrus. One major difference between their study and ours is that they used auditory input as a second modality instead of text. Further work is required to investigate to what extent the modality-invariant regions identified in our work generalize to all modalities.)

According to a range of theories, modality-invariant representations are tightly linked to (lexical-) semantic representations (*Simmons and Barsalou, 2003*; *Binder et al., 2009*; *Meschke and Gallant, 2024*). Most importantly, a range of studies that aimed to identify brain regions linked to semantic/conceptual representations by asking subjects to perform tasks that require semantic processing of words found evidence for such regions that overlap to a high degree with the regions identified in our study (*Fernandino et al., 2016*; *Martin et al., 2018*; *Carota et al., 2021*; *Fernandino et al., 2022*; *Tong et al., 2022*). A strong link between these systems could also explain our result that modality-agnostic decoders based on unimodal representations from language models are performing as well as decoders based on multimodal representations (*Figure 4*), as well as the partial left-lateralization of the identified modality-invariant regions.

The fact that the presence of modality-invariant patterns is positively correlated with the imagery decoding performance in different locations provides further evidence that the identified patterns are truly modality-invariant (they are not only shared between vision and language, but also for mental imagery). We further found that decoders trained exclusively on data for which participants were exposed to perceptual input to generalize to imagery trials, confirming previous findings that were based on more limited stimulus sets (*Stokes et al., 2009*; *Reddy et al., 2010*; *Lee et al., 2012*; *Johnson and Johnson, 2014*; *Naselaris et al., 2015*). Moreover, we found that modality-agnostic decoders outperform modality-specific decoders in terms of imagery decoding, demonstrating another advantage of this new type of decoder explicitly trained to pick up modality-invariant patterns.

The findings of our searchlight analysis for imagery decoding suggest that mental imagery indeed involves a large network of regions across both hemispheres of the cerebral cortex. This includes high-level visual areas, parietal areas, such as the precuneus and inferior parietal cortex and several frontal regions, but also parts of the early visual cortex. Results are highly similar on both hemispheres,

highlighting the involvement of large-scale bilateral brain networks during mental imagery of complex scenes.

While there are lesion studies on hemispheric asymmetries that suggest that regions in the left hemisphere are crucial for mental imagery (*Farah, 1984*; *Bartolomeo, 2002*), a more recent review that additionally considers evidence from neuroimaging and direct cortical stimulation studies suggests that frontoparietal networks in both hemispheres are involved in mental imagery, and that lateralization patterns can be found in the temporal lobes (*Liu et al., 2022*). Such lateralization was found to depend on the nature of the imagined items, the imagination of objects and words involving the left inferior temporal cortex, while the imagination of faces and people was found to be more right-lateralized, and the imagination of complex scenes (as in our study) led to significant activity in both hemispheres (*O'Craven and Kanwisher, 2000*; *Steel et al., 2021*; *Spagna et al., 2021*). Crucially, in more recent decoding studies, results were either observed bilaterally, or the analyses did not target hemispheric asymmetries (*Reddy et al., 2010*; *Lee et al., 2012*) (but see *Stokes et al., 2009* in which perception-to-imagery generalization of single letters was left-lateralized). Our work shows for the first time results of a searchlight analysis of imagery decoding of complex visual scenes. Above-chance decoding is possible on both hemispheres, with the highest decoding accuracies in the precuneus, the right inferior parietal cortex and the superior temporal sulcus. Future investigations with larger sets of imagined scenes could address the question of whether lateralization patterns depend on the nature of the imagined objects.

According to the results of our searchlight analysis, the anterior temporal lobes are not among the regions with the highest probability of being modality-invariant, contradicting the hypothesis of the hub-and-spoke theory that these areas are the major semantic hub in the brain. However, MRI signals from these regions have a lower signal-to-noise ratio with standard fMRI pulse sequences (*Devlin et al., 2000*; *Embleton et al., 2010*). A more targeted study with an adapted fMRI protocol would be required to shed light on the nature of patterns in these regions. More generally, the hub-and-spoke theory also puts emphasis on the role of spokes for the formation of conceptual representations (*Pobric et al., 2010*; *Ralph et al., 2017*). Future work could be aimed at testing the hub-and-spoke theory proposal that features in hierarchically lower level representation spaces of the spokes are not directly relatable to features in the representation space of the hubs: Object representations in the spokes are based on interpretable features (e.g. shape, color, affordances of an object) and get translated into another representational format that is representing conceptual similarities (but its dimensions do not directly map to interpretable features) in the semantic hub (*Frisby et al., 2023*). To test this hypothesis, modality-invariant representations in the anterior temporal lobes (measured with targeted fMRI pulse sequences) could be compared to representations in candidate regions for modality-specific spokes using RSA.

The modality-invariant regions we found in the searchlight analysis can also be seen as candidates for convergence zones, in which increasingly abstract representations are formed (*Damasio, 1989*; *Tranel et al., 1997*; *Meyer and Damasio, 2009*). To obtain further insight into the hierarchical organization of these zones, future work could take advantage of the improved temporal resolution of other brain imaging techniques, such as MEG to explore in which areas modality-invariant patterns are formed first, and how they are being transformed when spreading to higher-level areas of the brain (*Dirani and Pylkkänen, 2024*; *Benchetrit et al., 2024*).

In line with the GRAPES framework, (*Martin, 2009*; *Martin, 2016*), we found that modality-invariant representations are distributed across temporal and parietal areas. To test the related hypothesis that conceptual information is organized in domains, we plan to use RSA to understand which kind of semantic information is represented in the different modality-invariant regions identified.

Finally, our results can be interpreted with respect to the Global Workspace Theory. All modality-invariant regions are good candidate regions for a global workspace. They could, however, also be part of modality-specific modules that get activated in a modality-invariant fashion through a 'broadcast' operation as a stimulus is perceived consciously (*Baars, 1993*; *Baars, 2005*). To distinguish these two cases, an experimental manipulation of attention could be used: according to Global Workspace Theory, attention is required for information to enter the workspace, but not for the workspace signals to reach other brain regions via broadcast. In the future, we plan to investigate how modality-invariant patterns are modulated by attention, by analyzing additional test sessions from the same subjects in which they were instructed in specific runs to pay attention to

only one of the modalities (these sessions will be released as part of another future data set and publication).

To conclude, the results from our searchlight analysis so far are in line with all major theories on modality-invariant representations that were considered. As the dataset that this study was based on will be shared publicly, more targeted investigations can be performed by the research community in order to adjudicate between different theories.

## Limitations

There are some important limitations to this study, which are mainly consequences of some design choices to enable large-scale data collection. First, the fMRI experiment relied on a one-back task, which required subjects to remember preceding stimuli, and therefore, brain activity in each trial could be influenced by the previous stimuli kept in working memory. Such unrelated activity could have impaired the performance of decoders, which were only trained to decode the presented stimulus. However, this limitation could also represent an opportunity for future work on modality-invariant representations in working memory, for example, by training decoders to decode the preceding stimuli from such working memory activity. Second, the inclusion of the cross-modal one-back task could have incentivized subjects to perform visual imagery when reading the captions, or to recall descriptive words (and/or have inner speech) when viewing images. Such behavior might have facilitated the spread of modality-invariant information in the cortex. It remains an open question to what degree such widespread modality-invariant information is also present during isolated reading and visual inspection of images, without any cross-modal task.

Finally, it remains an open question whether the activation patterns in the modality-invariant regions identified in our study relate to abstract concepts or to lower-level features that are shared between the two modalities (see also Section Modality-invariant representations). Our current study did not aim to tackle this question, as it is not crucial for the goal of building modality-agnostic decoders. In the future, techniques, such as RSA and encoding models based on carefully designed feature spaces could be used to explore the nature of the representations in more detail. One major challenge to overcome will be the spatial overlap of modality-dependent, modality-invariant and abstract information in brain representations (*Fernandino et al., 2016*; *Liuzzi et al., 2020*; *Dirani and Pylkkänen, 2024*). More generally, *Binder, 2016* puts the dichotomy between modality-invariant and abstract representations into question, considering that 'there is no absolute demarcation between embodied/perceptual and abstract/conceptual representation in the brain.' (p. 1098). The author argues for a hierarchical system in which representational patterns become increasingly abstract, creating a continuum from actual experiential information up to higher-level conceptual information (see also *Andrews et al., 2014*).

## Acknowledgements

This research was funded by grants from the French Agence Nationale de la Recherche (ANR: AI-REPS grant number ANR-18-CE37-0007-01 and ANITI grant number ANR-19-PI3A-0004) as well as the European Union (ERC Advanced grant GLoW, 101096017). Views and opinions expressed are, however, those of the author(s) only and do not necessarily reflect those of the European Union or the European Research Council Executive Agency. Neither the European Union nor the granting authority can be held responsible for them. We thank the Inserm/UPS UMR1214 Technical Platform for their help in setting up and for the acquisitions of the MRI sequences.

## Additional information

### Funding

| Funder | Grant reference number | Author |
| --- | --- | --- |
| European Research Council | Advanced grant GLoW (101096017) | Rufin VanRullen |

| Funder | Grant reference number | Author |
|---|---|---|
| Agence Nationale de la Recherche | AI-REPS (ANR-18-CE37-0007-01) | Nicholas Asher Leila Reddy Rufin VanRullen |
| Agence Nationale de la Recherche | ANITI (ANR-19-PI3A-0004) | Nicholas Asher Leila Reddy Rufin VanRullen |

The funders had no role in study design, data collection and interpretation, or the decision to submit the work for publication.

## Author contributions

Mitja Nikolaus, Conceptualization, Resources, Data curation, Software, Formal analysis, Validation, Investigation, Visualization, Methodology, Writing – original draft, Project administration, Writing – review and editing; Milad Mozafari, Conceptualization, Resources, Data curation, Software, Formal analysis, Validation, Investigation, Methodology, Project administration; Isabelle Berry, Resources, Data curation; Nicholas Asher, Conceptualization, Funding acquisition, Methodology, Project administration; Leila Reddy, Conceptualization, Resources, Data curation, Formal analysis, Supervision, Funding acquisition, Validation, Investigation, Methodology, Project administration; Rufin VanRullen, Conceptualization, Data curation, Formal analysis, Supervision, Funding acquisition, Validation, Investigation, Methodology, Writing – original draft, Project administration, Writing – review and editing

## Author ORCIDs

Mitja Nikolaus ⬥ https://orcid.org/0000-0001-5609-6628
Milad Mozafari ⬥ https://orcid.org/0000-0002-4521-1640
Rufin VanRullen ⬥ https://orcid.org/0000-0002-3611-7716

## Ethics

Six subjects participated in the experiment after providing informed consent. The study was performed in accordance with French national ethical regulations (Comité de Protection des Personnes, ID 2019-A01920-57).

Reviewer #2 (Public review): https://doi.org/10.7554/eLife.107933.3.sa1
Reviewer #3 (Public review): https://doi.org/10.7554/eLife.107933.3.sa2
Author response https://doi.org/10.7554/eLife.107933.3.sa3

# Additional files

## Supplementary files

MDAR checklist

## Data availability

All fMRI data is available at https://openneuro.org/datasets/ds007272. Code for preprocessing and analyses is available at https://github.com/mitjanikolaus/multimodal_decoding (copy archived at *Nikolaus, 2026*).

The following dataset was generated:

| Author(s) | Year | Dataset title | Dataset URL | Database and Identifier |
|---|---|---|---|---|
| Nikolaus M, Mozafari M, Berry I, Asher N, Reddy L, VanRullen R | 2025 | SemReps-8K | https://doi.org/10.18112/openneuro.ds007272.v1.0.0 | OpenNeuro, 10.18112/openneuro.ds007272.v1.0.0 |

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

# Appendix 1

## Dataset quality metrics
### Head motion

*Appendix 1—figure 1* and *Appendix 1—figure 2* present information on head motion for each subject. All head motion estimates are based on the realignment parameters calculated using SPM12. We found that head motion estimates are largely stable and do not vary extensively between sessions and subjects.

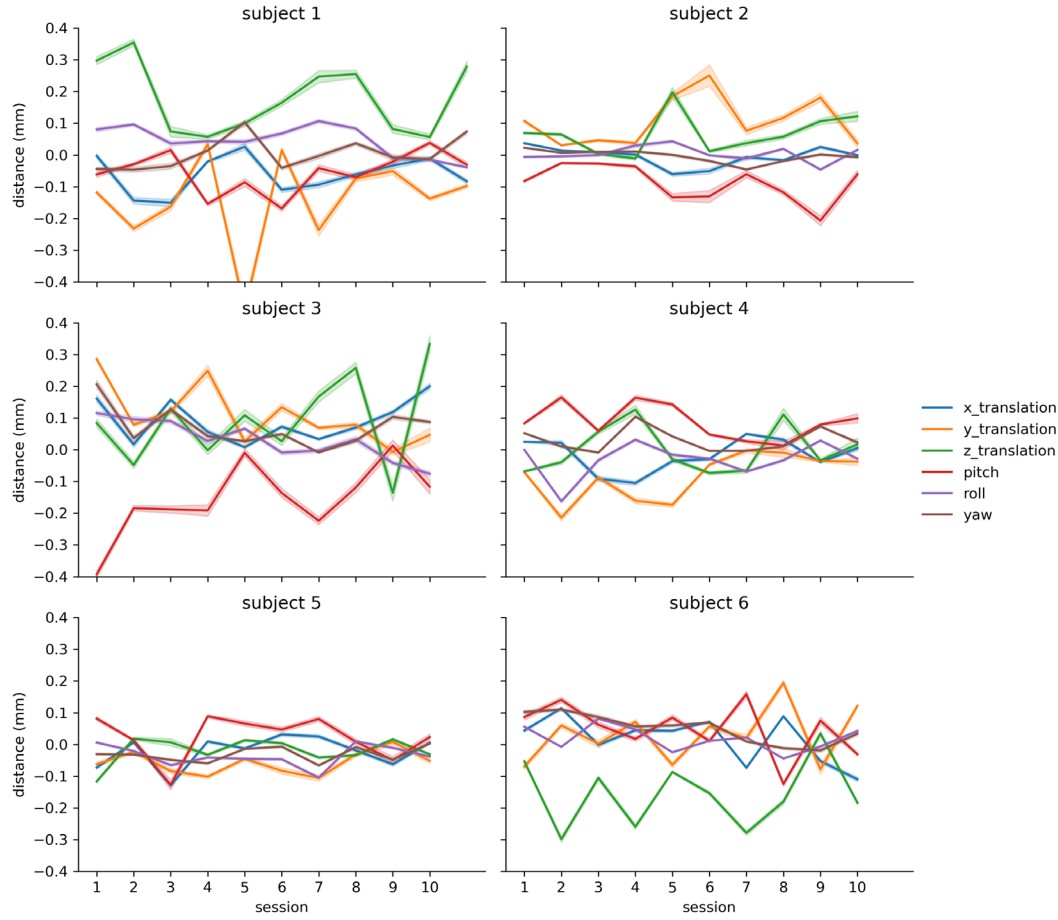

**Appendix 1—figure 1.** Head motion estimates for each subject. The plots show the realignment parameters as computed by SPM12 (spm_realign) for estimating within-modality rigid body alignment. We multiplied the rotation parameters pitch, roll, and yaw (originally in radian units) by 50 in order to allow interpretation in terms of millimeters of displacement for a circle of diameter 10 cm (which is approximately the mean distance from the cerebral cortex to the center of the head) (*Power et al., 2012*). The translucent error bands show 95% confidence intervals calculated using bootstrapping over all frames/runs from a session.

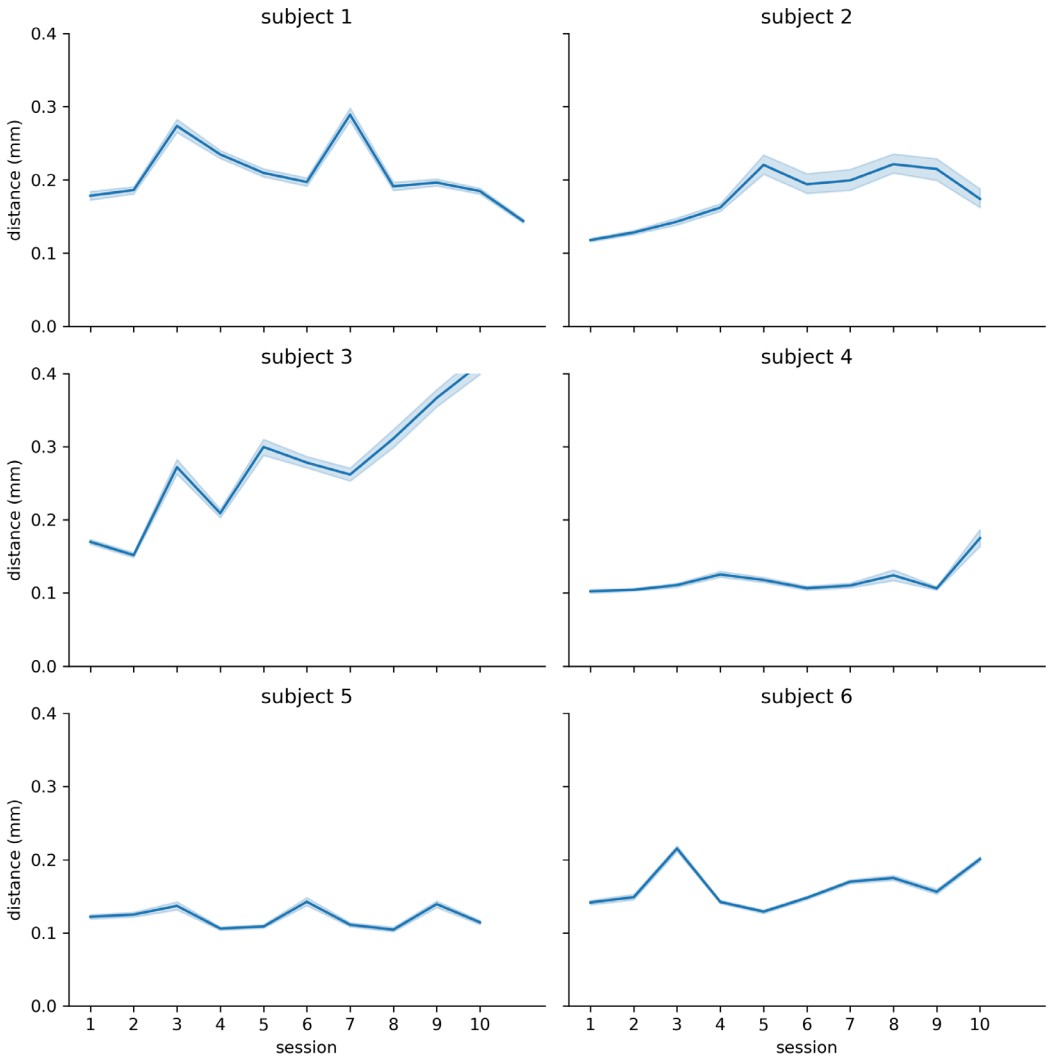

**Appendix 1—figure 2.** Frame-wise displacement for each subject. The measure indicates how much the head changed position from one frame to the next. We calculated framewise displacement as the sum of the absolute values of the derivatives of the six realignment parameters. The translucent error bands show 95% confidence intervals calculated using bootstrapping over all frames/runs from a session.

## Intersession alignment

During preprocessing, all functional MRI data was coregistered to the anatomical scan of the first session. To assess to which degree the alignment of functional data of different sessions varies with respect to the alignment of the first session, we estimated mutual information between the coregistered scans and the reference anatomical scan. During coregistration, our pipeline based on SPM12 created a mean image of the functional data for each session to estimate transformation parameters specific to each scanning session. We calculated mutual information between the coregistered mean images and the anatomical scan of the first session. Then, we normalized all values with respect to the mutual information of the first session (i.e. the normalized mutual information value for session 1 for all subjects is 1). The results are shown in *Appendix 1—figure 3*. The plot shows that the mutual information does not deviate strongly from the baseline (minimum value: 90%) and stays in an interval of ±5% for most subjects.

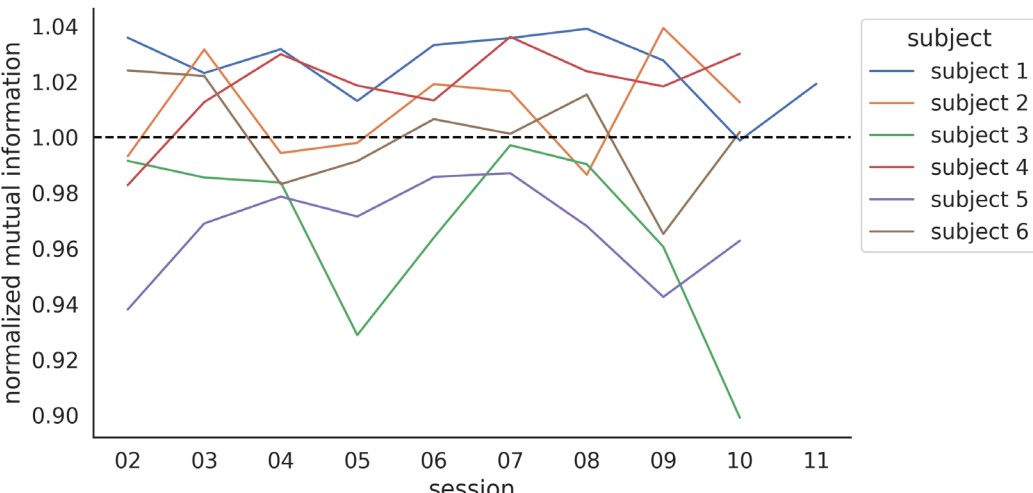

**Appendix 1—figure 3.** Mutual information between the anatomical scan of the first session and functional data for each session as an indicator of intersession alignment. All values were normalized based on the mutual information from the functional data from the first session. The raw mutual information scores for the alignment during the first session are: subject 1: 0.49; subject 2: 0.55; subject 3: 0.55; subject 4: 0.53; subject 5: 0.57; subject 6: 0.57.

## Appendix 2

### Feature extraction details

For each target stimulus (image or caption), our database also contained an equivalent stimulus in the other modality (caption or image). In this way, we could extract model features from the corresponding image for vision models, the corresponding caption for language models, and a multimodal representation of both image and caption for the multimodal models. We used publicly available pretrained models implemented in the Hugging Face Transformers library (**Wolf et al., 2020**) or from their respective authors' repositories.

Model versions for unimodal models are as indicated in **Figure 4**. For multimodal models, the exact version for CLIP was `clip-vit-large-patch14`, for ViLT `vilt-b32-mlm`, for VisualBERT `visualbert-nlvr2-coco-pre`, for ImageBind `imagebind_huge`, for Bridgetower `bridgetower-large-itm-mlm-itc`, for Flava `flava-full`, for SigLip `siglip-so400m-patch14-384`, and for Paligemma2 `paligemma2-3b-pt-224`.

We extracted language features from all models by averaging the outputs for each token, as this was established as common practice for the extraction of sentence embeddings from Transformer-based language models (**Krasnowska-Kieraś and Wróblewska, 2019**; **Reimers and Gurevych, 2019**).

For Transformer-based vision models, we compared representations extracted by averaging the outputs for each patch with representations extracted from [CLS] tokens in **Appendix 2—table 1**. We found that for almost all models, the mean features allow for higher decoding accuracies. For all experiments reported in the main paper, we, therefore, only considered this method.

**Appendix 2—table 1.** Feature comparison for vision models.
Pairwise accuracy for modality-agnostic decoders based on vision features extracted by averaging the last hidden states of all patches ('vision_features_mean') compared to when using features extracted from [CLS] tokens ('vision_features_cls'). The method leading to the best decoding performance for each model is highlighted in bold.

| Model | Vision features | Pairwise accuracy |
| --- | --- | --- |
| dino-base | vision_features_cls | 0.763 |
| **dino-base** | **vision_features_mean** | **0.819** |
| dino-giant | vision_features_cls | 0.750 |
| **dino-giant** | **vision_features_mean** | **0.820** |
| dino-large | vision_features_cls | 0.754 |
| **dino-large** | **vision_features_mean** | **0.816** |
| **vit-b-16** | **vision_features_cls** | **0.788** |
| vit-b-16 | vision_features_mean | 0.772 |
| vit-h-14 | vision_features_cls | 0.785 |
| **vit-h-14** | **vision_features_mean** | **0.804** |
| vit-l-16 | vision_features_cls | 0.788 |
| **vit-l-16** | **vision_features_mean** | **0.796** |

For multimodal models, we also compared a range of techniques for feature extraction. The results are shown in **Appendix 2—table 2**.

**Appendix 2—table 2.** Feature comparison for multimodal models.
The features are either based on the [CLS] tokens from the fused representations ('fused_cls'), averaging over the fused tokens ('fused_mean'), or averaging over tokens from intermediate vision and language stream outputs ('avg'). For the last case, for some models, the vision and language features can also be either based on [CLS] or based on averaging over all tokens/patches. The method leading to the best decoding performance for each model is highlighted in bold.

| Model | Features | Vision features | Language features | Pairwise accuracy |
|---|---|---|---|---|
| blip2 | avg | vision_features_cls | lang_features_cls | 0.851 |
| blip2 | fused_cls | vision_features_cls | lang_features_cls | 0.710 |
| blip2 | fused_mean | vision_features_cls | lang_features_cls | 0.740 |
| bridgetower | fused_cls | | | 0.815 |
| bridgetower | fused_mean | | | 0.788 |
| clip | avg | vision_features_cls | lang_features_cls | 0.842 |
| flava | avg | vision_features_cls | lang_features_cls | 0.842 |
| flava | fused_cls | vision_features_cls | lang_features_cls | 0.752 |
| flava | fused_mean | vision_features_cls | lang_features_cls | 0.772 |
| imagebind | avg | vision_features_cls | lang_features_cls | 0.857 |
| paligemma2 | avg | vision_features_cls | lang_features_mean | 0.829 |
| paligemma2 | avg | vision_features_mean | lang_features_mean | 0.848 |
| paligemma2 | fused_mean | vision_features_mean | lang_features_mean | 0.828 |
| siglip | avg | vision_features_cls | lang_features_cls | 0.852 |
| siglip | avg | vision_features_mean | lang_features_cls | 0.823 |
| vilt | fused_cls | | | 0.759 |
| vilt | fused_mean | | | 0.839 |
| visualbert | fused_cls | | | 0.639 |
| visualbert | fused_mean | | | 0.743 |

For dual-stream multimodal models, we averaged the vision and language features to create the final multimodal feature representation. For single-stream multimodal features, we compared using representations extracted by averaging the outputs for each token with representations extracted from [CLS] token and found that the averaged output leads to better performance in almost all cases (see *Appendix 2—table 2*). Furthermore, some single-stream models (Flava, Paligemma2, and BLIP2) allow for feature extraction based on intermediate vision and language representations in addition to a direct extraction of multimodal features (based on fused representations from the multimodal stream). We found that averaging features from these intermediate vision and language representations leads to better performance (see *Appendix 2—table 2*). For all results reported in the main paper, we used the feature extraction method leading to best performance for each model.

## Decoder training details

The decoders were linear ridge-regression models as implemented in the scikit-learn library (*Pedregosa et al., 2011*). All training data was standardized to have a mean of 0 and a standard deviation of 1. The test data was standardized using the mean and standard deviation of the training data. The regularization hyperparameter $\alpha$ was optimized using fivefold cross-validation on the training set (values considered: $\alpha \in \{1e3, 1e4, 1e5, 1e6, 1e7\}$). Afterwards, a final model was trained using the best $\alpha$ on the whole training set.

## Appendix 3

### Qualitative decoding results for modality-specific decoders

In this section, we present qualitative decoding results for modality-specific decoders for the same examples as presented in Section Qualitative Decoding Results for modality-agnostic decoders. The results show that modality-agnostic decoders are as good as modality-specific decoders when evaluated in a within-modality decoding setup (*Appendix 3—figure 1* and *Appendix 3—figure 3*) but substantially better than modality-specific decoders when evaluated in a cross-decoding setup (*Appendix 3—figures 2 and 4*).

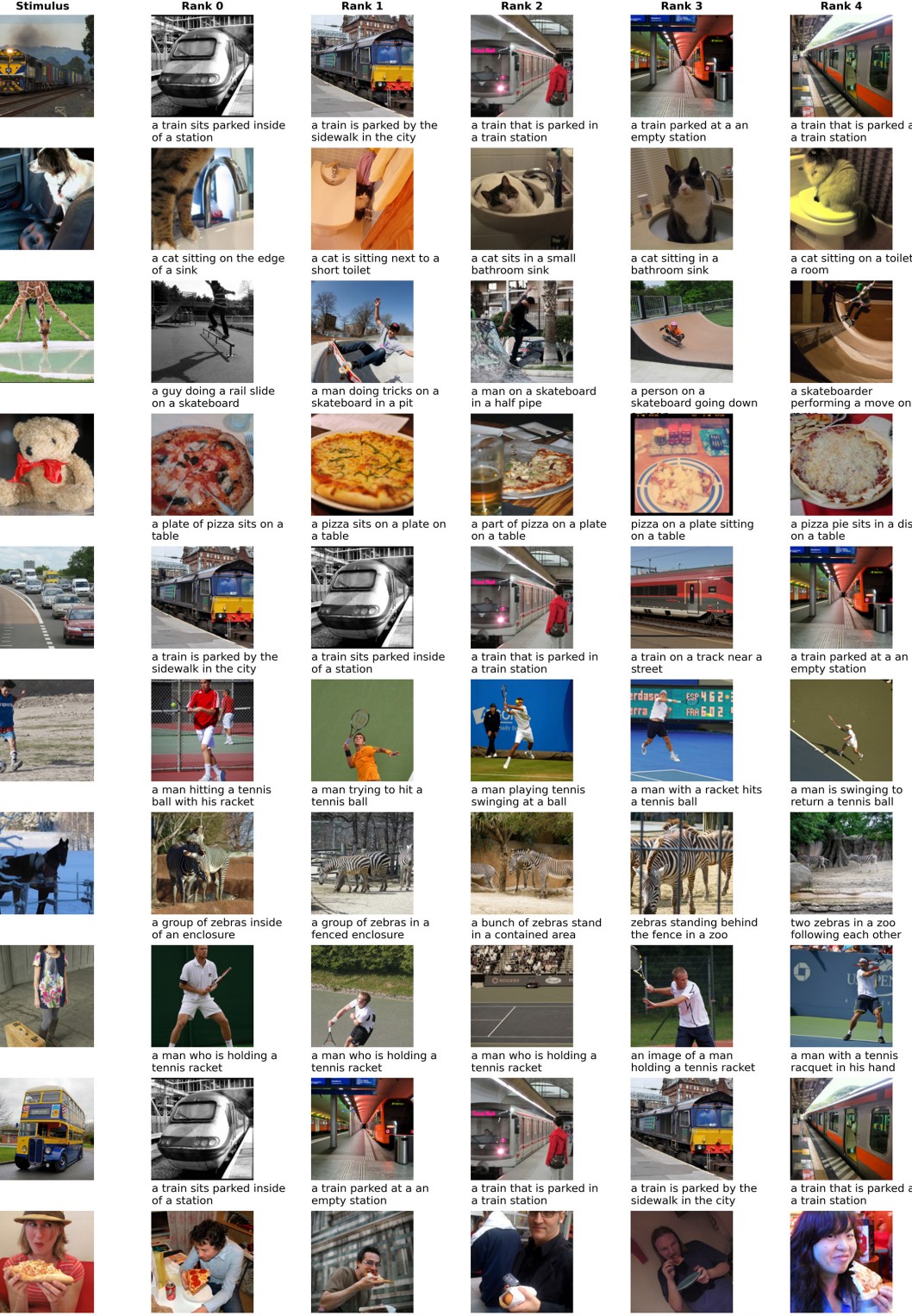

**Appendix 3—figure 1.** Decoding examples for image decoding using a modality-specific decoder trained on images (within-modality decoding). The first column shows the image the subject was seeing and the five following columns show the candidate stimuli with highest similarity to the predicted features, in descending order. We display both the image and the caption of the candidate stimuli because the decoder is based on multimodal features that are extracted from both modalities. All images were taken from the CoCo dataset (*Lin et al., 2014*).

| | Stimulus | Rank 0 | Rank 1 | Rank 2 | Rank 3 | Rank 4 |
|---|---|---|---|---|---|---|

**Appendix 3—figure 2.** Decoding examples for caption decoding using a modality-specific decoder trained on images (cross-modality decoding). For details, see caption of *Figure 1* All images were taken from the CoCo dataset (*Lin et al., 2014*).

| Stimulus | Rank 0 | Rank 1 | Rank 2 | Rank 3 | Rank 4 |
|---|---|---|---|---|---|
| a train coming up the tracks through trees | a train pulls into a station out side | a train pulls up to a train stop | a passenger train pulls into a train station | a passenger train pulled up to a train stop | a train slows down to a stop at the station |
| a small dog standing inside a car | a cat sitting on the corner of a bed | a cat and a large clock in a room | a large suitcase on the ground in a room | a cat laying on a bed in a room | a dog standing on the side of a road |
| a giraffe drinking water from a man-made pond | a family of elephants eating hay in their pen | three giraffes standing next to a feed box | three elephants standing together by a fence | herd of elephants in an enclosure at the zoo | three giraffes stand in a row in a pen |
| the teddy bear has a big bright red bow | a rusty hydrant with weeds at its base | a yummy looking hot dog with catsup on it | a closeup picture of a polar bears head | a yellow fire hydrant mostly covered in rust | an old, rusting, yellow fire hydrant n weeds |
| a lot of cars driving on a packed highway | street signs are posted along a busy street | a busy street with some cars driving by | some traffic turning left on a busy street | a number of cars on a city street with signs | a large street with hardly any drivers in it |
| a young boy playing soccer on a dirt field | a lady and a man getting ready to fly a kite | a boy holding a frisbee walks beside a girl | two children plays with a kite in the field | two men in a field prepare to fly kites | two men play a game of frisbee in a field |
| two horses standing near a fence in the snow | two park benches on a snowy grass feild | a couple of zebras graze on some grass | a couple of zebras graze on some grass | two zebras grazing grass in a secluded spot | two zebras wander a tall grassy plains area |
| a woman stands waiting next to her suitcase | a man playing wii while others watch or wait | a woman is watching a kid and man playing wii | two people playing wii with an onlooker | a man plays wii while others look on | two men standing by each other playing wii |
| a double decker bus is in the street | a train is traveling on the railroad alone | a train that is on the tracks near snow | a train is driving on the rails of a track | a train is driving on the train tracks | a train operating on a set of train tracks |
| a woman eating a big piece of pizza | a woman biting a pizza while a man holds it | two men stand eating large slices of pizza | two people eating a couple of hot dogs | a man and a woman are ready to eat a pizza | two people enjoying a pizza from a tray |

**Appendix 3—figure 3.** Decoding examples for caption decoding using a modality-specific decoder trained on captions (within-modality decoding). For details, see caption of *Appendix 3—figure 1* All images were taken from the CoCo dataset (*Lin et al., 2014*).

| Stimulus | Rank 0 | Rank 1 | Rank 2 | Rank 3 | Rank 4 |
|---|---|---|---|---|---|

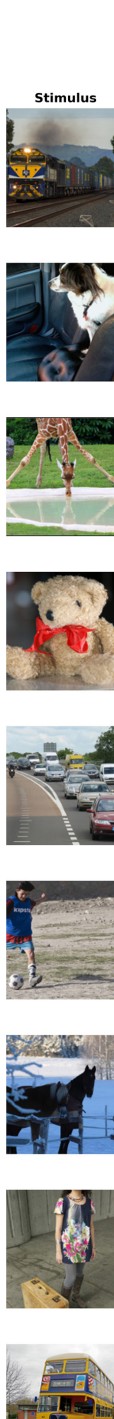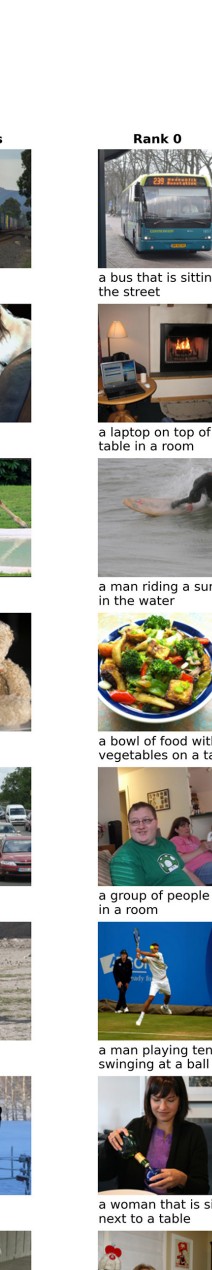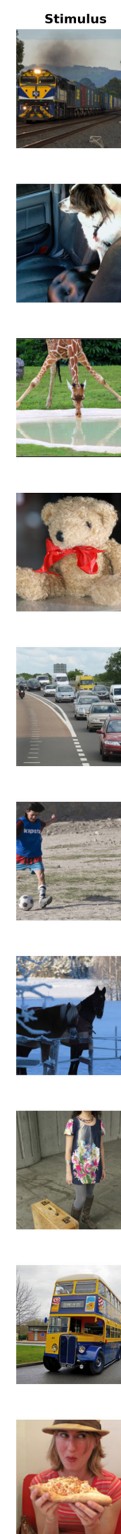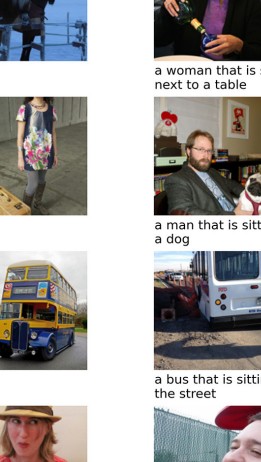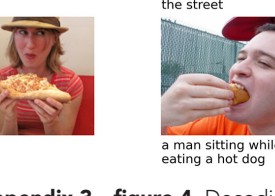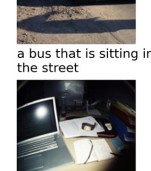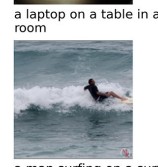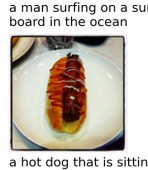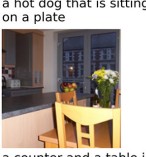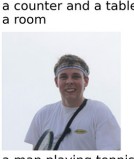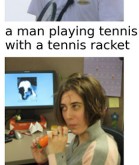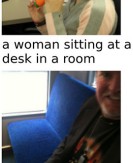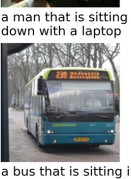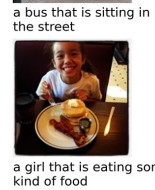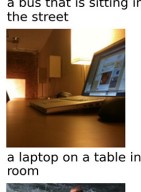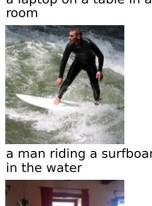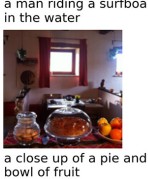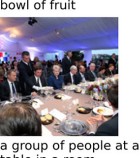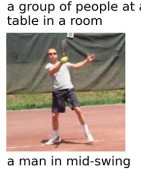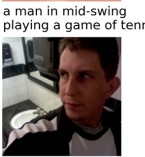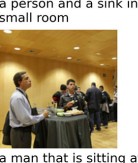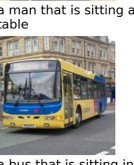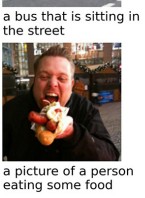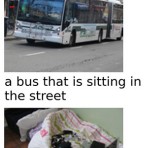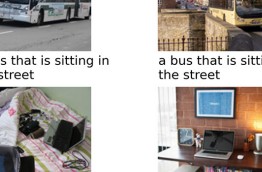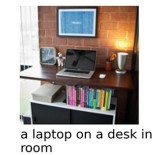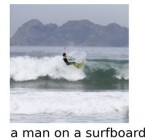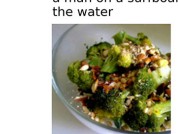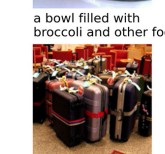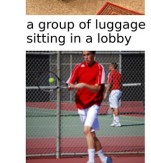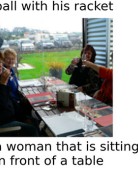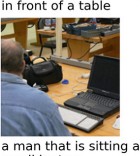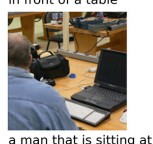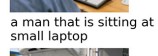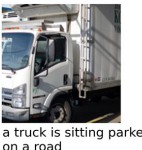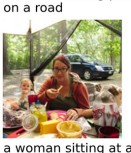

**Appendix 3—figure 4.** Decoding examples for image decoding using a modality-specific decoder trained on captions (cross-modality decoding). For details, see caption of *Appendix 3—figure 1* All images were taken from the CoCo dataset (*Lin et al., 2014*).

# Appendix 4

## Candidates for modality-invariant regions

*Appendix 4—table 1* shows candidates for modality-invariant regions that were identified by previous work. We only considered fMRI experiments involving multiple stimulus modalities. Many other studies relied on unimodal stimuli and semantic tasks to identify modality-invariant regions; these are, however, less directly comparable to our setup.

**Appendix 4—table 1.** Candidates for modality-invariant regions, as identified by previous work. All these regions were also found in our analysis, except for the two regions marked with an asterisk (*).

| Hemi | Region | Studies that identified the region as modality-invariant |
|---|---|---|
| | Superior occipital gyrus* | *Vandenberghe et al., 1996*; *Shinkareva et al., 2011* |
| | Middle occipital gyrus | *Shinkareva et al., 2011* |
| | Inferior occipital gyrus | *Shinkareva et al., 2011*; *Simanova et al., 2014* |
| | Superior temporal gyrus | *Shinkareva et al., 2011* |
| | Superior temporal sulcus | *Man et al., 2012* |
| | Middle temporal gyrus | *Vandenberghe et al., 1996*; *Shinkareva et al., 2011*; *Fairhall and Caramazza, 2013*; *Devereux et al., 2013*; *Simanova et al., 2014*; *Handjaras et al., 2016* |
| | Inferior temporal gyrus | *Vandenberghe et al., 1996*; *Shinkareva et al., 2011*; *Fairhall and Caramazza, 2013*; *Simanova et al., 2014*; *Handjaras et al., 2016* |
| | Fusiform gyrus | *Vandenberghe et al., 1996*; *Moore and Price, 1999*; *Bright et al., 2004*; *Shinkareva et al., 2011*; *Fairhall and Caramazza, 2013*; *Simanova et al., 2014* |
| | Hippocampus | *Vandenberghe et al., 1996* |
| | Parahippocampus | *Bright et al., 2004*; *Fairhall and Caramazza, 2013*; *Handjaras et al., 2016* |
| | Perirhinal cortex | *Bright et al., 2004*; *Fairhall and Caramazza, 2013* |
| | Superior parietal cortex | *Shinkareva et al., 2011* |
| | Inferior parietal cortex | *Shinkareva et al., 2011*; *Handjaras et al., 2016* |
| | Cingulate gyrus | *Moore and Price, 1999*; *Fairhall and Caramazza, 2013*; *Handjaras et al., 2016* |
| | Cuneus | *Shinkareva et al., 2011* |
| | Precuneus | *Shinkareva et al., 2011*; *Handjaras et al., 2016*; *Fairhall and Caramazza, 2013*; *Popham et al., 2021* |
| Left hemisphere | Supramarginal gyrus | *Shinkareva et al., 2011*; *Handjaras et al., 2016* |

*Appendix 4—table 1 Continued on next page*

*Appendix 4—table 1 Continued*

| Hemi | Region | Studies that identified the region as modality-invariant |
|---|---|---|
| | Angular gyrus | *Shinkareva et al., 2011*; *Fairhall and Caramazza, 2013*; *Devereux et al., 2013*; *Simanova et al., 2014*; *Handjaras et al., 2016*; *Popham et al., 2021* |
| | Intraparietal sulcus | *Shinkareva et al., 2011*; *Devereux et al., 2013* |
| | Temporoparietal junction | *Vandenberghe et al., 1996*; *Handjaras et al., 2016* |
| | Superior frontal gyrus | *Fairhall and Caramazza, 2013* |
| | Middle frontal gyrus | *Fairhall and Caramazza, 2013*; *Handjaras et al., 2016* |
| | Inferior frontal gyrus | *Vandenberghe et al., 1996*; *Bright et al., 2004*; *Simanova et al., 2014*; *Liuzzi et al., 2017*; *Handjaras et al., 2016* |
| | Precentral gyrus | *Shinkareva et al., 2011* |
| | Postcentral gyrus | *Shinkareva et al., 2011* |
| | Paracentral lobule | *Shinkareva et al., 2011* |
| | Supplementary motor area | *Shinkareva et al., 2011* |
| | Fusiform gyrus | *Shinkareva et al., 2011*; *Simanova et al., 2014* |
| | Superior temporal sulcus | *Man et al., 2012* |
| | Middle temporal gyrus | *Handjaras et al., 2016* |
| | Inferior temporal gyrus | *Handjaras et al., 2016* |
| | Angular gyrus | *Handjaras et al., 2016*; *Popham et al., 2021* |
| | Superior parietal cortex | *Shinkareva et al., 2011* |
| | Precuneus | *Shinkareva et al., 2011*; *Popham et al., 2021*; *Fairhall and Caramazza, 2013*; *Handjaras et al., 2016* |
| | Cingulate gyrus | *Moore and Price, 1999*; *Fairhall and Caramazza, 2013*; *Handjaras et al., 2016*; *Jung et al., 2018* |
| | Parahippocampus | *Handjaras et al., 2016* |
| | Inferior parietal cortex | *Handjaras et al., 2016* |
| | Paracentral lobule | *Shinkareva et al., 2011* |
| | Middle frontal gyrus | *Simanova et al., 2014*; *Jung et al., 2018* |
| | Superior frontal gyrus* | *Jung et al., 2018* |
| Right hemisphere | Inferior frontal gyrus | *Moore and Price, 1999*; *Simanova et al., 2014* |

# Appendix 5

## Searchlight size

In order to optimize the size of the searchlight for this analysis, we first ran a searchlight analysis with a fixed radius of 10 mm using a modality-agnostic decoder on the data from the first subject (sub-01). Due to the shape of the cortex, this leads of searchlights that contain varying numbers of vertices (on average: 897.4; max: 1580; min: 399). By observing the decoding scores as a function of the number of vertices, we find that performance peaks at 750 vertices (*Appendix 5—table 1*). The final searchlight analysis was, therefore, performed with a searchlight of a fixed number of 750 vertices. The average radius with this number of vertices was 9.41 mm (max: 13.65 mm).

**Appendix 5—table 1.** Average decoding scores (for images and captions) by number of vertices. Scores were calculated based on the results of a searchlight analysis with a radius of 10 mm. The accuracy values were grouped into bins based on the number of vertices that the searchlight was based on.

| # Vertices | Pairwise accuracy (mean) |
| --- | --- |
| 250 | 51.81% |
| 500 | 52.35% |
| 750 | 55.96% |
| 1000 | 55.49% |
| 1250 | 54.60% |
| 1500 | 52.96% |

## Appendix 6

### Per-subject results

Results for individual subjects can be found in *Appendix 6—figure 1*. Among all subjects, we found similar converging results for decoding accuracies when comparing models, feature modalities, and modality-agnostic with modality-specific decoders.

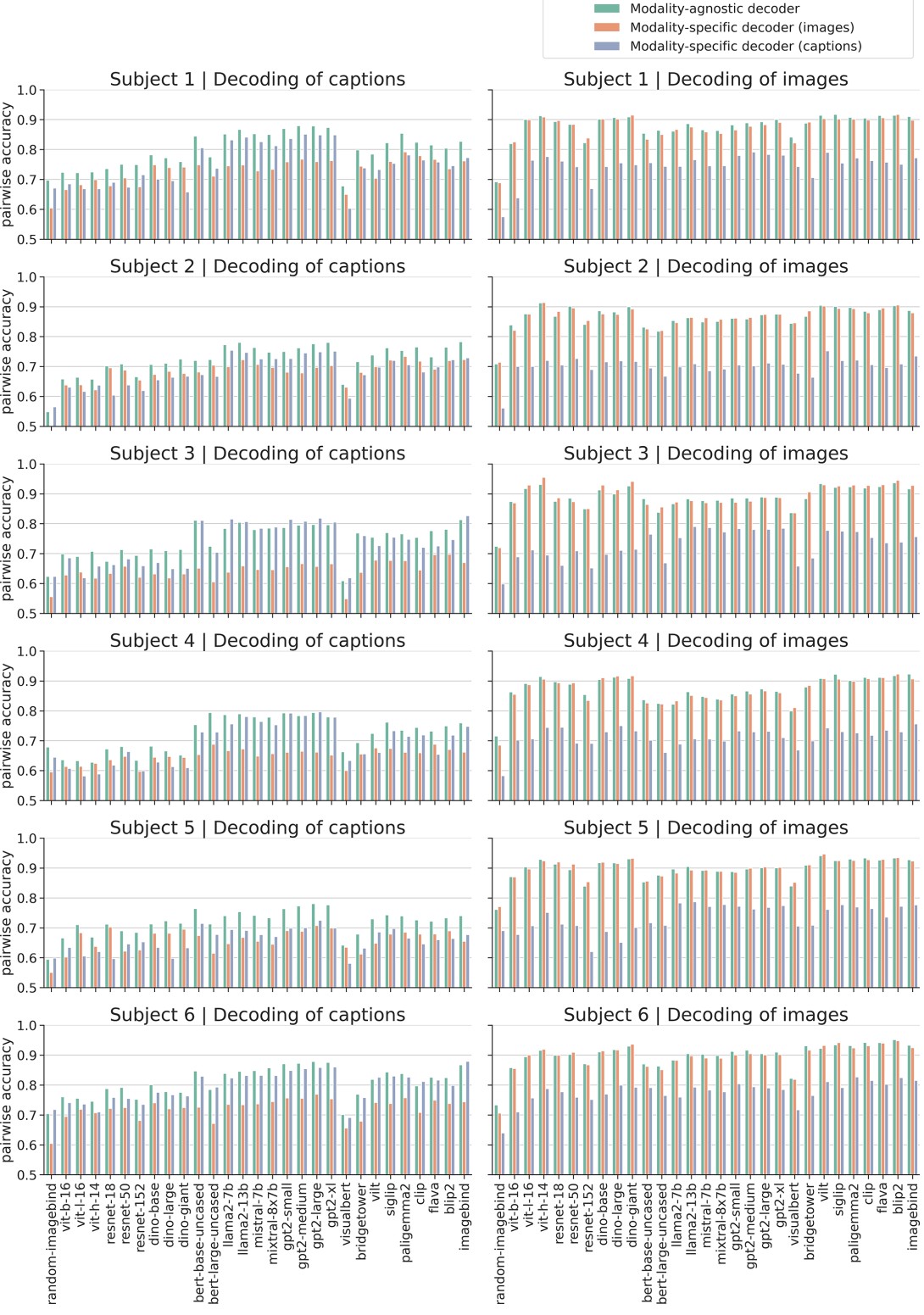

**Appendix 6—figure 1.** Pairwise accuracy per subject. For details, refer to *Figure 5*

## Appendix 7

## Qualitative imagery decoding results

The following *Appendix 7—figures 1–6* present qualitative decoding results for the imagery conditions using a modality-agnostic decoder. We present the results separately for each subject as each subject chose an individual set of three stimuli to perform mental imagery on. In each plot, the leftmost column shows the caption that was used as initial instruction as well as the subject's sketch that they drew at the end of the experiment.

We used the same large candidate set of 41 K stimuli as for the qualitative decoding results for the other conditions (see also Section Qualitative Decoding Results).

Overall, we found that the decoding quality for imagery stimuli lags behind that for trials with perceived stimuli. This was expected and confirms the quantitative results reported in Section Imagery Decoding. Still, in several cases, some of the concepts are decoded correctly (e.g. a woman in the first row of *Appendix 7—figure 1*; winter sports in the second row of *Appendix 7—figure 3*; laptops/screens and multiple people in the third row of *Appendix 7—figure 3*).

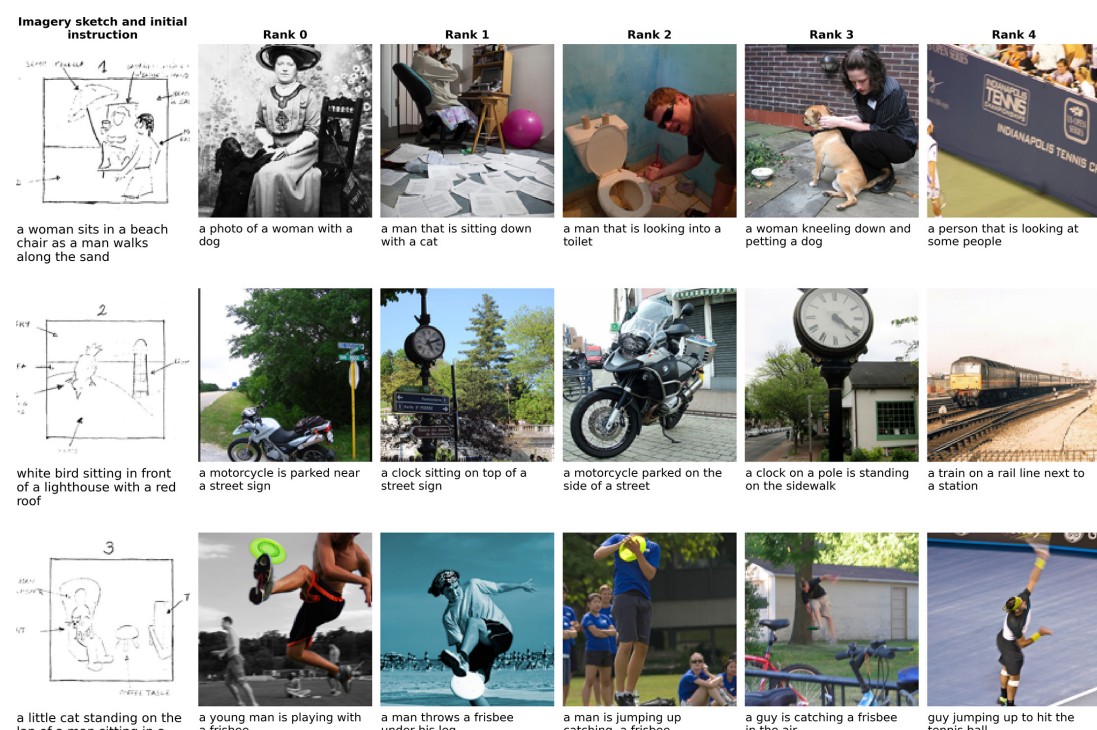

**Appendix 7—figure 1.** Imagery decoding for Subject 1 using a modality-agnostic decoder. The first column shows the caption that was used to stimulate imagery and on top of it the sketch of their mental image that the subjects drew at the end of the experiment. The five following columns show the candidate stimuli with highest similarity to the predicted features, in descending order. We display both the image and the caption of the candidate stimuli because the decoder is based on multimodal features that are extracted from both modalities. All images were taken from the CoCo dataset (*Lin et al., 2014*).

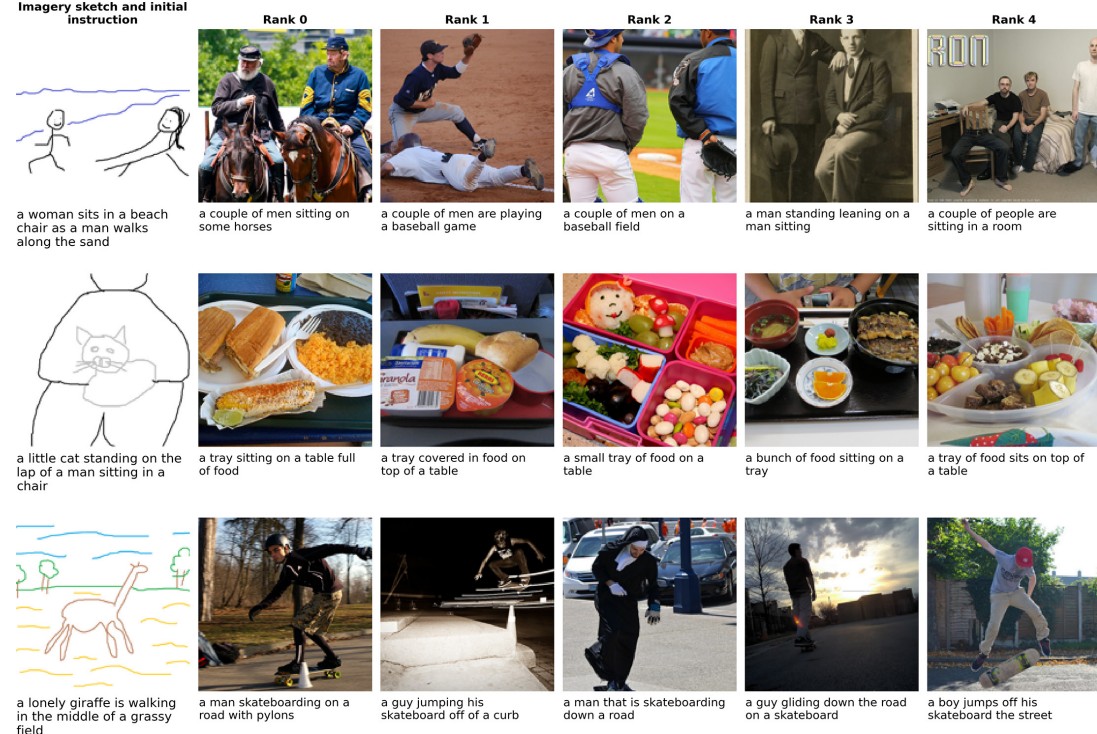

**Appendix 7—figure 2.** Imagery decoding for Subject 2 using a modality-agnostic decoder. For further details, refer to the caption of *Appendix 7—figure 1*. All images were taken from the CoCo dataset (*Lin et al., 2014*).

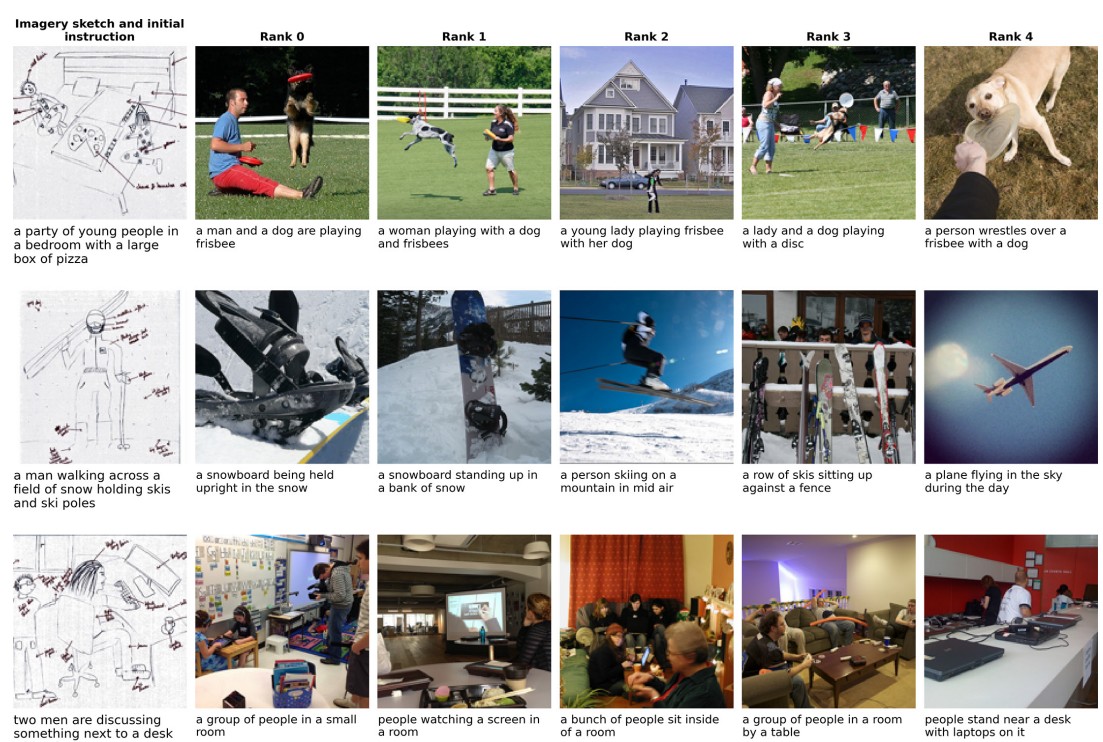

**Appendix 7—figure 3.** Imagery decoding for Subject 3 using a modality-agnostic decoder. For further details, refer to the caption of *Appendix 7—figure 1*. All images were taken from the CoCo dataset (*Lin et al., 2014*).

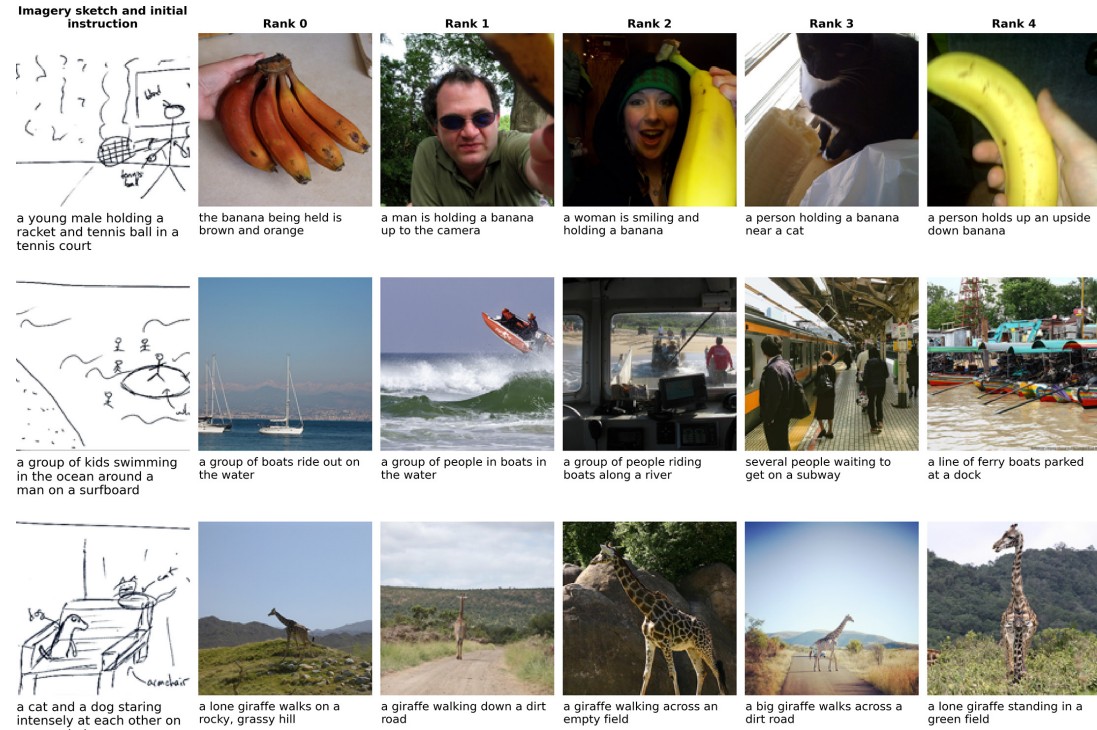

**Appendix 7—figure 4.** Imagery decoding for Subject 4 using a modality-agnostic decoder. For further details, refer to the caption of *Appendix 7—figure 1*. All images were taken from the CoCo dataset (*Lin et al., 2014*).

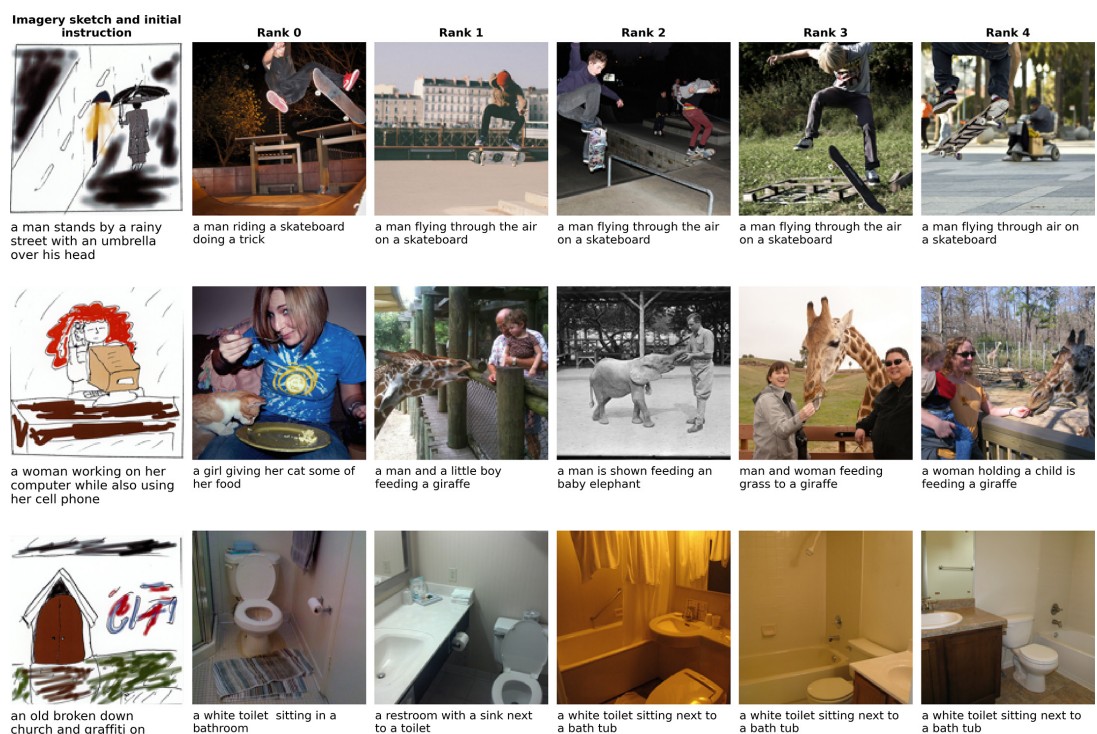

**Appendix 7—figure 5.** Imagery decoding for Subject 5 using a modality-agnostic decoder. For further details, refer to the caption of *Appendix 7—figure 1*. All images were taken from the CoCo dataset (*Lin et al., 2014*).

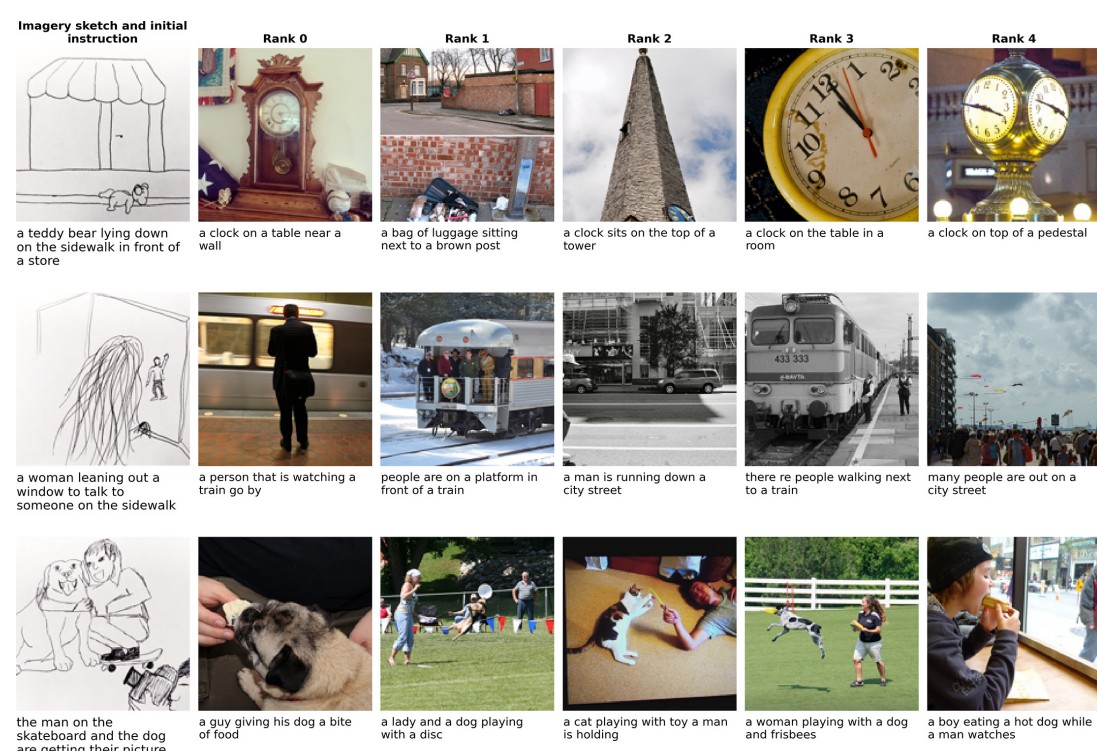

**Appendix 7—figure 6.** Imagery decoding for Subject 6 using a modality-agnostic decoder. For further details, refer to the caption of *Appendix 7—figure 1*. All images were taken from the CoCo dataset (*Lin et al., 2014*).

