## [Editor Report · eLife Assessment]

The study introduces a **valuable** dataset for investigating the relationship between vision and language in the brain. The authors provide **convincing** evidence that decoders trained on brain responses to both images and captions outperform those trained on responses to a single modality. The dataset and decoder results will be of interest to communities studying brain and machine decoding.

---

## [Referee Report · Reviewer #2 (Public review)]

Summary:

This work presents a modality-agnostic decoder trained on a large fMRI dataset (SemReps-8K), in which subjects viewed natural images and corresponding captions. The decoder predicts stimulus content from brain activity irrespective of the input modality and performs on par with-or even outperforms-modality-specific decoders. Its success depends more on the diversity of brain data (multimodal vs. unimodal) than on whether the feature-extraction models are visual, linguistic, or multimodal. Particularly, the decoder shows strong performance in decoding imagery content. These results suggest that the modality-agnostic decoder effectively leverages shared brain information across image and caption tasks.

Strengths:

(1) The modality-agnostic decoder compellingly leverages multimodal brain information, improving decoding accuracy-particularly for non-sensory input such as captions-showing high methodological and application value.

(2) The dataset is a substantial and well-controlled contribution, with >8,000 image-caption trials per subject and careful matching of stimuli across modalities-an essential resource for testing theories about different representational modalities.

Weakness:

In the searchlight analysis aimed at identifying modality-invariant representations, although the combined use of four decoding conditions represents a relatively strict approach, the underlying logic remains unclear. The modality-agnostic decoder has demonstrated strong sensitivity in decoding brain activity, as shown earlier in the paper, whereas the cross-decoding with modality-specific decoders is inherently more conservative. If, as the authors note, the modality-agnostic decoder might have learned to leverage different features to project stimuli from different modalities, then taking the union of conditions would seem more appropriate. Conversely, if the goal is to obtain a more conservative result, why not focus solely on the cross-decoding conditions? The relationships among the four decoding conditions are not clearly delineated, and the contrasts between them might themselves yield valuable insights. As it stands, however, the logic of the current approach is not straightforward.

---

## [Referee Report · Reviewer #3 (Public review)]

Summary:

The authors recorded brain responses while participants viewed images and captions. The images and captions were taken from the COCO dataset, so each image has a corresponding caption and each caption has a corresponding image. This enabled the authors to extract features from either the presented stimulus or the corresponding stimulus in the other modality. The authors trained linear decoders to take brain responses and predict stimulus features. "Modality-specific" decoders were trained on brain responses to either images or captions while "modality-agnostic" decoders were trained on brain responses to both stimulus modalities. The decoders were evaluated on brain responses while the participants viewed and imagined new stimuli, and prediction performance was quantified using pairwise accuracy. The authors reported the following results:

(1) Decoders trained on brain responses to both images and captions can predict new brain responses to either modality.

(2) Decoders trained on brain responses to both images and captions outperform decoders trained on brain responses to a single modality.

(3) Many cortical regions represent the same concepts in vision and language.

(4) Decoders trained on brain responses to both images and captions can decode brain responses to imagined scenes.

Strengths:

This is an interesting study that addresses important questions about modality-agnostic representations. Previous work has shown that decoders trained on brain responses to one modality can be used to decode brain responses to another modality. The authors build on these findings by collecting a new multimodal dataset and training decoders on brain responses to both modalities.

To my knowledge, SemReps-8K is the first dataset of brain responses to vision and language where each stimulus item has a corresponding stimulus item in the other modality. This means that brain responses to a stimulus item can be modeled using visual features of the image, linguistic features of the caption, or multimodal features derived from both the image and the caption. The authors also employed a multimodal one-back matching task which forces the participants to activate modality-agnostic representations. Overall, SemReps-8K is a valuable resource that will help researchers answer more questions about modality-agnostic representations.

The analyses are also very comprehensive. The authors trained decoders on brain responses to images, captions, and both modalities, and they tested the decoders on brain responses to images, caption, and imagined scenes. They extracted stimulus features using a range of visual, linguistic, and multimodal models. The modeling framework appears rigorous and the results offer new insights into the relationship between vision, language, and imagery. In particular, the authors found that decoders trained on brain responses to both images and captions were more effective at decoding brain responses to imagined scenes than decoders trained on brain responses to either modality in isolation. The authors also found that imagined scenes can be decoded from a broad network of cortical regions.

Weaknesses:

The characterization of "modality-agnostic" and "modality-specific" decoders seems a bit contradictory. There are three major choices when fitting a decoder: the modality of the training stimuli, the modality of the testing stimuli, and the model used to extract stimulus features. However, the authors characterize their decoders based on only the first choice-"modality-specific" decoders were trained on brain responses to either images or captions while "modality-agnostic" decoders were trained on brain responses to both stimulus modalities. I think that this leads to some instances where the conclusions are inconsistent with the methods and results.

First, the authors suggest that "modality-specific decoders are not explicitly encouraged to pick up on modality-agnostic features during training" (line 137) while "modality-agnostic decoders may be more likely to leverage representations that are modality-agnostic" (line 140). However, whether a decoder is required to learn modality-agnostic representations depends on both the training responses and the stimulus features. Consider the case where the stimuli are represented using linguistic features of the captions. When you train a "modality-specific" decoder on image responses, the decoder is forced to rely on modality-agnostic information that is shared between the image responses and the caption features. On the other hand, when you train a "modality-agnostic" decoder on both image responses and caption responses, the decoder has access to the modality-specific information that is shared by the caption responses and the caption features, so it is not explicitly required to learn modality-agnostic features. As a result, while the authors show that "modality-agnostic" decoders outperform "modality-specific" decoders in most conditions, I am not convinced that this is because they are forced to learn more modality-agnostic features.

Second, the authors claim that "modality-specific decoders can be applied only in the modality that they were trained on" while "modality-agnostic decoders can be applied to decode stimuli from multiple modalities, even without knowing a priori the modality the stimulus was presented in" (line 47). While "modality-agnostic" decoders do outperform "modality-specific" decoders in the cross-modality conditions, it is important to note that "modality-specific" decoders still perform better than expected by chance (figure 5). It is also important to note that knowing about the input modality still improves decoding performance even for "modality-agnostic" decoders, since it determines the optimal feature space-it is better to decode brain responses to images using decoders trained on image features, and it is better to decode brain responses to captions using decoders trained on caption features.

Comments on revised version:

The revised version benefits from clearer claims and more precise terminology (i.e. classifying the decoders as "modality-agnostic" or "modality-specific" while classifying the representations as "modality-invariant" or "modality-dependent").

While the modality-agnostic decoders outperform the modality-specific decoders, I am still not convinced that this is because they are "explicitly trained to leverage the shared information in modality-invariant patterns of the brain activity". On one hand, the high-level feature spaces may each contain some amount of modality-invariant information, so even modality-specific decoders can capture some modality-invariant information. On the other hand, I do not see how training the modality-agnostic decoders on responses to both modalities necessitates that they learn modality-invariant representations beyond those that are learned by the modality-specific decoders.

---

## [Author Response]

The following is the authors’ response to the original reviews

We would like to thank all reviewers for their constructive and in-depth reviews. Thanks to your feedback, we realized that the main objective of the paper was not presented clearly enough, and that our use of the same “modality-agnostic” terminology for both decoders and representations caused confusion. We addressed these two major points as outlined in the following.

In the revised manuscript, we highlight that the main contribution of this paper is to introduce modality-agnostic decoders. Apart from introducing this new decoder type, we put forward their advantages in comparison to modality-specific decoders in terms of decoding performance and analyze the modality-invariant representations (cf. updated terminology in the following paragraph) that these decoders rely on. The dataset that these analyses are based on is released as part of this paper, in the spirit of open science (but this dataset is only a secondary contribution for our paper).

Regarding the terminology, we clearly define modality-agnostic decoders as decoders that are trained on brain imaging data from subjects exposed to stimuli in multiple modalities. The decoder is not given any information on which modality a stimulus was presented in, and is therefore trained to operate in a modality-agnostic way. In contrast, modality-specific decoders are trained only on data from a single stimulus modality. These terms are explained in Figure 2. While these terms describe different ways of how decoders can be trained, there are also different ways to evaluate them afterwards (see also Figure 3); but obviously, this test-time evaluation does not change the nature of the decoder, i.e., there is no contradiction in applying a modality-specific decoder to brain data from a different modality.

Further, we identify representations that are relevant for modality-agnostic decoders using the searchlight analysis. We realized that our choice of using the same “modality-agnostic” term to describe these brain representations created unnecessary debate and confusion. In order to not conflate the terminology, in the updated manuscript we call these representations modality-invariant (and the opposite modality-dependent). Our methodology does not allow us to distinguish whether certain representations merely share representational structure to a certain degree, or are truly representations that abstract away from any modality-dependent information. However, in order to be useful for modality-agnostic decoding, a significant degree of shared representational structure is sufficient, and it is this property of brain representations that we now define as “modality-invariant”.

We updated the manuscript in line with this new terminology and focus: in particular, the first Related Work section on Modality-invariant brain representations, as well as the Introduction and Discussion.

**Public Reviews:**

**Reviewer #1 (Public review):**
Summary:The authors introduce a densely-sampled dataset where 6 participants viewed images and sentence descriptions derived from the MS Coco database over the course of 10 scanning sessions. The authors further showcase how image and sentence decoders can be used to predict which images or descriptions were seen, using pairwise decoding across a set of 120 test images. The authors find decodable information widely distributed across the brain, with a left-lateralized focus. The results further showed that modality-agnostic models generally outperformed modality-specific models, and that data based on captions was not explained better by caption-based models but by modality-agnostic models. Finally, the authors decoded imagined scenes.Strengths:(1) The dataset presents a potentially very valuable resource for investigating visual and semantic representations and their interplay.(2) The introduction and discussion are very well written in the context of trying to understand the nature of multimodal representations and present a comprehensive and very useful review of the current literature on the topic.Weaknesses:(1) The paper is framed as presenting a dataset, yet most of it revolves around the presentation of findings in relation to what the authors call modality-agnostic representations, and in part around mental imagery. This makes it very difficult to assess the manuscript, whether the authors have achieved their aims, and whether the results support the conclusions.

Thanks for this insightful remark. The dataset release is only a secondary contribution of our study; this was not clear enough in the previous version. We updated the manuscript to make the main objective of the paper more clear, as outlined in our general response to the reviews (see above).

(2) While the authors have presented a potential use case for such a dataset, there is currently far too little detail regarding data quality metrics expected from the introduction of similar datasets, including the absence of head-motion estimates, quality of intersession alignment, or noise ceilings of all individuals.

As already mentioned in the general response, the main focus of the paper is to introduce modality-agnostic decoders. The dataset is released in addition, this is why we did not focus on reporting extensive quality metrics in the original manuscript. To respond to your request, we updated the appendix of the manuscript to include a range of data quality metrics.

The updated appendix includes head motion estimates in the form of realignment parameters and framewise displacement, as well as a metric to assess the quality of intersession alignment. More detailed descriptions can be found in Appendix 1 of the updated manuscript.

Estimating noise ceilings based on repeated presentations of stimuli (as for example done in Allen et al. (2022)) requires multiple betas for each stimulus. All training stimuli were only presented once, so this could only be done for the test stimuli which were presented repeatedly. However, during our preprocessing procedure we directly calculated stimulus-specific betas based on data from all sessions using one single GLM, which means that we did not obtain separate betas for repeated presentations of the same stimulus. We will however share the raw data publicly, so that such noise ceilings can be calculated using an adapted preprocessing procedure if required.

Allen, E. J., St-Yves, G., Wu, Y., Breedlove, J. L., Prince, J. S., Dowdle, L. T., Nau, M., Caron, B., Pestilli, F., Charest, I., Hutchinson, J. B., Naselaris, T., & Kay, K. (2022). A massive 7T fMRI dataset to bridge cognitive neuroscience and artificial intelligence. Nature Neuroscience, 25(1), 116–126. https://doi.org/10.1038/s41593-021-00962-x

(3) The exact methods and statistical analyses used are still opaque, making it hard for a reader to understand how the authors achieved their results. More detail in the manuscript would be helpful, specifically regarding the exact statistical procedures, what tests were performed across, or how data were pooled across participants.

In the updated manuscript, we improved the level of detail for the descriptions of statistical analyses wherever possible (see also our response to your “Recommendations for the authors”, Point 6).

Regarding data pooling across participants:

Figure 8 shows averaged results across all subjects (as indicated in the caption)

Regarding data pooling for the estimation of the significance threshold of the searchlight analysis for modality-invariant regions: We updated the manuscript to clarify that we performed a permutation test, combined with a bootstrapping procedure to estimate a group-level null distribution: “For each subject, we evaluated the decoders 100 times with shuffled labels to create per-subject chance-level results. Then, we randomly selected one of the 100 chance-level results for each of the 6 subjects and calculated group-level statistics (TFCE values) the exact same way as described in the preceding paragraph. We repeated this procedure 10,000 times resulting in 10,000 permuted group-level results.”

Additionally, we indicated that the same permutation testing methods were applied to assess the significance threshold for the imagery decoding searchlight maps (Figure 10).

(4) Many findings (e.g., Figure 6) are still qualitative but could be supported by quantitative measures.

The Figures 6 and 7 are intentionally qualitative results to support the quantitative decoding results presented in Figures 4 and 5. (see also Reviewer 2 Comment 2)

Figures 4 and 5 show pairwise decoding accuracy as a quantitative measure for evaluation of the decoders. This metric is the main metric we used to compare different decoder types and features. Based on the finding that modality-agnostic decoders using imagebind features achieve the best score on this metric, we performed the additional qualitative analysis presented in Figures 6 and 7. (Note that we expanded the candidate set for the qualitative analysis in order to have a larger and more diverse set of images.)

(5) Results are significant in regions that typically lack responses to visual stimuli, indicating potential bias in the classifier. This is relevant for the interpretation of the findings. A classification approach less sensitive to outliers (e.g., 70-way classification) could avoid this issue. Given the extreme collinearity of the experimental design, regressors in close temporal proximity will be highly similar, which could lead to leakage effects.

It is true that our searchlight analysis revealed significant activity in regions outside of the visual cortex. However, it is assumed that the processing of visual information does not stop at the border of the visual cortex. The integration of information such as the semantics of the image is progressively processed in other higher-level regions of the brain. Recent studies have shown that activity in large areas of the cortex (including many outside of the visual cortex) can be related to visual stimulation (Solomon et al. 2024; Raugel et al. 2025). Our work confirms this finding and we therefore do not see reason to believe that this is due to a bias in our decoders.

Further, you are suggesting that we could replace our regression approach with a 70-way classification. However, this is difficult using our fMRI data as we do not see a straightforward way to assign the training and testing stimuli with class labels (the two datasets consist of non-overlapping sets of naturalistic images).

To address your concerns regarding the collinearity of the experimental design and possible leakage effects, we trained and evaluated a decoder for one subject after running a “null-hypothesis” adapted preprocessing. More specifically, for all sessions, we shifted the functional data of all runs by one run (moving the data of the last run to the very front), but leaving the design matrices in place. Thereby, we destroyed the relationship of stimuli and brain activity but kept the original data and design with its collinearity (and possible biases). We preprocessed this adapted data for subject 1, and ran a whole-brain decoding using Imagebind features and verified that the decoding performance was at chance level: Pairwise accuracy (captions): 0.43 | Pairwise accuracy (images): 0.47 | Pairwise accuracy (imagery): 0.50. This result provides evidence against the notion that potential collinearity or biases in our experimental design or evaluation procedure could have led to inflated results.

Raugel, J., Szafraniec, M., Vo, H.V., Couprie, C., Labatut, P., Bojanowski, P., Wyart, V. and King, J.R. (2025). Disentangling the Factors of Convergence between Brains and Computer Vision Models. arXiv preprint arXiv:2508.18226.

Solomon, S. H., Kay, K., & Schapiro, A. C. (2024). Semantic plasticity across timescales in the human brain. bioRxiv, 2024-02.

(6) The manuscript currently lacks a limitations section, specifically regarding the design of the experiment. This involves the use of the overly homogenous dataset Coco, which invites overfitting, the mixing of sentence descriptions and visual images, which invites imagery of previously seen content, and the use of a 1-back task, which can lead to carry-over effects to the subsequent trial.

Regarding the dataset CoCo: We agree that CoCo is somewhat homogenous, it is however much more diverse and naturalistic than the smaller datasets used in previous fMRI experiments with multimodal stimuli. Additionally, CoCo has been widely adopted as a benchmark dataset in the Machine Learning community, and features rich annotations for each image (e.g. object labels, segmentations, additional captions, people’s keypoints) facilitating many more future analyses based on our data.

Regarding the mixing of sentence descriptions and images: Subjects were not asked to visualize sentences and different techniques for the one-back tasks might have been used. Generally, we do not see it as problematic if subjects are performing visual imagery to some degree while reading sentences, and this might even be the case during normal reading as well. A more targeted experiment comparing reading with and without interleaved visual stimulation in the form of images and a one-back task would be required to assess this, but this was not the focus of our study. For now, it is true that we can not be sure that our results generalize to cases in which subjects are just reading and are less incentivized to perform mental imagery.

Regarding the use of a 1-back task: It was necessary to make some design choices in order to realize this large-scale data collection with approximately 10 hours of recording per subject. Specifically, the 1-back task was included in the experimental setup in order to assure continuous engagement of the participant during the rather long sessions of 1 hour. The subjects did indeed need to remember the previous stimulus to succeed at the 1-back task, which means that some brain activity during the presentation of a stimulus is likely to be related to the previous stimulus. We aimed to account for this confound during the preprocessing stage when fitting the GLM, which was fit to capture only the response to the presented image/caption, not the preceding one. Still, it might have picked up on some of the activity from preceding stimuli, causing some decrease of the final decoding performance.

We added a limitations section to the updated manuscript to discuss these important issues.

(7) I would urge the authors to clarify whether the primary aim is the introduction of a dataset and showing the use of it, or whether it is the set of results presented. This includes the title of this manuscript. While the decoding approach is very interesting and potentially very valuable, I believe that the results in the current form are rather descriptive, and I'm wondering what specifically they add beyond what is known from other related work. This includes imagery-related results. This is completely fine! It just highlights that a stronger framing as a dataset is probably advantageous for improving the significance of this work.

Thanks a lot for pointing this out. Based on this comment and feedback from the other reviewers we restructured the abstract, introduction and discussion section of the paper to better reflect the primary aim. (cf. general response above).

You further mention that it is not clear what our results add beyond what is known from related work. We list the main contributions here:

A single modality-agnostic decoder can decode the semantics of visual and linguistic stimuli irrespective of the presentation modality with a performance that is not lagging behind modality-specific decoders.

Modality-agnostic decoders outperform modality-specific decoders for decoding captions and mental imagery.

Modality-invariant representations are widespread across the cortex a range of previous work has suggested they were much more localized (Bright et al. 2004; Jung et al. 2018; Man et al. 2012; Simanova et al. 2014).

Regions that are useful for imagery are largely overlapping with modality-invariant regions

Bright, P., Moss, H., & Tyler, L. K. (2004). Unitary vs multiple semantics: PET studies of word and picture processing. Brain and language, 89(3), 417-432.

Jung, Y., Larsen, B., & Walther, D. B. (2018). Modality-Independent Coding of Scene Categories in Prefrontal Cortex. Journal of Neuroscience, 38(26), 5969–5981.

Liuzzi, A. G., Bruffaerts, R., Peeters, R., Adamczuk, K., Keuleers, E., De Deyne, S., Storms, G., Dupont, P., & Vandenberghe, R. (2017). Cross-modal representation of spoken and written word meaning in left pars triangularis. NeuroImage, 150, 292–307. https://doi.org/10.1016/j.neuroimage.2017.02.032

Man, K., Kaplan, J. T., Damasio, A., & Meyer, K. (2012). Sight and Sound Converge to Form Modality-Invariant Representations in Temporoparietal Cortex. Journal of Neuroscience, 32(47), 16629–16636.

Simanova, I., Hagoort, P., Oostenveld, R., & van Gerven, M. A. J. (2014). Modality-Independent Decoding of Semantic Information from the Human Brain. Cerebral Cortex, 24(2), 426–434.

**Reviewer #2 (Public review):**
Summary:This study introduces SemReps-8K, a large multimodal fMRI dataset collected while subjects viewed natural images and matched captions, and performed mental imagery based on textual cues. The authors aim to train modality-agnostic decoders--models that can predict neural representations independently of the input modality - and use these models to identify brain regions containing modality-agnostic information. They find that such decoders perform comparably or better than modality-specific decoders and generalize to imagery trials.Strengths:(1) The dataset is a substantial and well-controlled contribution, with >8,000 image-caption trials per subject and careful matching of stimuli across modalities - an essential resource for testing theories of abstract and amodal representation.(2) The authors systematically compare unimodal, multimodal, and cross-modal decoders using a wide range of deep learning models, demonstrating thoughtful experimental design and thorough benchmarking.(3) Their decoding pipeline is rigorous, with informative performance metrics and whole-brain searchlight analyses, offering valuable insights into the cortical distribution of shared representations.(4) Extension to mental imagery decoding is a strong addition, aligning with theoretical predictions about the overlap between perception and imagery.Weaknesses:While the decoding results are robust, several critical limitations prevent the current findings from conclusively demonstrating truly modality-agnostic representations:(1) Shared decoding ≠ abstraction: Successful decoding across modalities does not necessarily imply abstraction or modality-agnostic coding. Participants may engage in modality-specific processes (e.g., visual imagery when reading, inner speech when viewing images) that produce overlapping neural patterns. The analyses do not clearly disambiguate shared representational structure from genuinely modality-independent representations. Furthermore, in Figure 5, the modality-agnostic encoder did not perform better than the modality-specific decoder trained on images (in decoding images), but outperformed the modality-specific decoder trained on captions (in decoding captions). This asymmetry contradicts the premise of a truly "modality-agnostic" encoder. Additionally, given the similar performance between modality-agnostic decoders based on multimodal versus unimodal features, it remains unclear why neural representations did not preferentially align with multimodal features if they were truly modality-independent.

We agree that successful modality-agnostic and cross-modal decoding does not necessarily imply that abstract patterns were decoded. In the updated manuscript, we therefore refer to these representations as modality-invariant (see also the updated terminology explained in the general response above).

If participants are performing mental imagery when reading, and this is allowing us to perform cross-decoding, then this means that modality-invariant representations are formed during this mental imagery process, i.e. that the representations formed during this form of mental imagery are compatible with representations during visual perception (or, in your words, produce overlapping neural patterns). While we can not know to what extent people were performing mental imagery while reading (or having inner speech while viewing images), our results demonstrate that their brain activity allows for decoding across modalities, which implies that modality-invariant representations are present.

It is true that our current analyses can not disambiguate modality-invariant representations (or, in your words, shared representational structure) from abstract representations (in your words, genuinely modality-independent representations). As the main goal of the paper was to build modality-agnostic decoders, and these only require what we call “modality-invariant” representations (see our updated terminology in the general reviewer response above), we leave this question open for future work. We do however discuss this important limitation in the Discussion section of the updated manuscript.

Regarding the asymmetry of decoding results when comparing modality-agnostic decoders with the two respective modality-specific decoders for captions and images: We do not believe that this asymmetry contradicts the premise of a modality-agnostic decoder. Multiple explanations for this result are possible: (1) The modality-specific decoder for images might benefit from the more readily decodable lower-level modality-dependent neural activity patterns in response to images, which are less useful for the modality-agnostic decoder because they are not useful for decoding caption trials. The modality-specific decoders for captions might not be able to pick up on low-level modality-dependent neural activity patterns as these might be less easily decodable.

The signal-to-noise ratio for caption trials might be lower than for image trials (cf. generally lower caption decoding performance), therefore the addition of training data (even if it is from another modality) improves the decoding performance for captions, but not for images (which might be at ceiling already).

Regarding the similar performance between modality-agnostic decoders based on multimodal versus unimodal features: Unimodal features are based on rather high-level features of the respective modality (e.g. last-layer features of a model trained for semantic image classification), which can be already modality-invariant to some degree. Additionally, as already mentioned before, in the updated manuscript we only require representations to be modality-invariant and not necessarily abstract.

(2) The current analysis cannot definitively conclude that the decoder itself is modality-agnostic, making "Qualitative Decoding Results" difficult to interpret in this context. This section currently provides illustrative examples, but lacks systematic quantitative analyses.

The qualitative decoding results in Figures 6 and 7 present exemplary qualitative results for the quantitative results presented in Figures 4 and 5 (see also Reviewer 1 Comment 4).

Figures 4 and 5 show pairwise decoding accuracy as a quantitative measure for evaluation of the decoders. This metric is the main metric we used to compare different decoder types and features. Based on the finding that modality-agnostic decoders using imagebind features achieve the best score on this metric, we performed the additional qualitative analysis presented in Figures 6 and 7. (Note that we expanded the candidate set for the qualitative analysis in order to have a larger and more diverse set of images.)

(3) The use of mental imagery as evidence for modality-agnostic decoding is problematic.

Imagery involves subjective, variable experiences and likely draws on semantic and perceptual networks in flexible ways. Strong decoding in imagery trials could reflect semantic overlap or task strategies rather than evidence of abstraction.

It is true that mental imagery does not necessarily rely on modality-agnostic representations. In the updated manuscript we revised our terminology and refer to the analyzed representations as modality-invariant, which we define as “representations that significantly overlap between modalities”.

The manuscript presents a methodologically sophisticated and timely investigation into shared neural representations across modalities. However, the current evidence does not clearly distinguish between shared semantics, overlapping unimodal processes, and true modality-independent representations. A more cautious interpretation is warranted.Nonetheless, the dataset and methodological framework represent a valuable resource for the field.

We fully agree with these observations, and updated our terminology as outlined in the general response.

**Reviewer #3 (Public review):**
Summary:The authors recorded brain responses while participants viewed images and captions. The images and captions were taken from the COCO dataset, so each image has a corresponding caption, and each caption has a corresponding image. This enabled the authors to extract features from either the presented stimulus or the corresponding stimulus in the other modality.The authors trained linear decoders to take brain responses and predict stimulus features."Modality-specific" decoders were trained on brain responses to either images or captions, while "modality-agnostic" decoders were trained on brain responses to both stimulus modalities. The decoders were evaluated on brain responses while the participants viewed and imagined new stimuli, and prediction performance was quantified using pairwise accuracy. The authors reported the following results:(1) Decoders trained on brain responses to both images and captions can predict new brain responses to either modality.(2) Decoders trained on brain responses to both images and captions outperform decoders trained on brain responses to a single modality.(3) Many cortical regions represent the same concepts in vision and language.(4) Decoders trained on brain responses to both images and captions can decode brain responses to imagined scenes.Strengths:This is an interesting study that addresses important questions about modality-agnostic representations. Previous work has shown that decoders trained on brain responses to one modality can be used to decode brain responses to another modality. The authors build on these findings by collecting a new multimodal dataset and training decoders on brain responses to both modalities.To my knowledge, SemReps-8K is the first dataset of brain responses to vision and language where each stimulus item has a corresponding stimulus item in the other modality. This means that brain responses to a stimulus item can be modeled using visual features of the image, linguistic features of the caption, or multimodal features derived from both the image and the caption. The authors also employed a multimodal one-back matching task, which forces the participants to activate modality-agnostic representations. Overall, SemReps-8K is a valuable resource that will help researchers answer more questions about modality-agnostic representations.The analyses are also very comprehensive. The authors trained decoders on brain responses to images, captions, and both modalities, and they tested the decoders on brain responses to images, captions, and imagined scenes. They extracted stimulus features using a range of visual, linguistic, and multimodal models. The modeling framework appears rigorous, and the results offer new insights into the relationship between vision, language, and imagery. In particular, the authors found that decoders trained on brain responses to both images and captions were more effective at decoding brain responses to imagined scenes than decoders trained on brain responses to either modality in isolation. The authors also found that imagined scenes can be decoded from a broad network of cortical regions.Weaknesses:The characterization of "modality-agnostic" and "modality-specific" decoders seems a bit contradictory. There are three major choices when fitting a decoder: the modality of the training stimuli, the modality of the testing stimuli, and the model used to extract stimulus features. However, the authors characterize their decoders based on only the first choice-"modality-specific" decoders were trained on brain responses to either images or captions, while "modality-agnostic" decoders were trained on brain responses to both stimulus modalities. I think that this leads to some instances where the conclusions are inconsistent with the methods and results.

In our analysis setup, a decoder is entirely determined by two factors: (1) the modality of the stimuli that the subject was exposed to, and (2) the machine learning model used to extract stimulus features.

The modality of the testing stimuli defines whether we are evaluating the decoder in a within-modality or cross-modality setting, but is not an inherent characteristic of a trained decoder

First, the authors suggest that "modality-specific decoders are not explicitly encouraged to pick up on modality-agnostic features during training" (line 137) while "modality-agnostic decoders may be more likely to leverage representations that are modality-agnostic" (line 140). However, whether a decoder is required to learn modality-agnostic representations depends on both the training responses and the stimulus features. Consider the case where the stimuli are represented using linguistic features of the captions. When you train a "modality-specific" decoder on image responses, the decoder is forced to rely on modality-agnostic information that is shared between the image responses and the caption features. On the other hand, when you train a "modality-agnostic" decoder on both image responses and caption responses, the decoder has access to the modality-specific information that is shared by the caption responses and the caption features, so it is not explicitly required to learn modality-agnostic features. As a result, while the authors show that "modality-agnostic" decoders outperform "modality-specific" decoders in most conditions, I am not convinced that this is because they are forced to learn more modality-agnostic features.

It is true that for example a modality-specific decoder trained on fmri data from images with stimulus features extracted from captions might also rely on modality-invariant features. We still call this decoder modality-specific, as it has been trained to decode brain activity recorded from a specific stimulus modality. In the updated manuscript we corrected the statement that “modality-specific decoders are not explicitly encouraged to pick up on modality-invariant features during training” to include the case of decoders trained on features from the other modality which might also rely on modality-invariant features.

It is true that a modality-agnostic decoder can also have access to modality-dependent information for captions and images. However, as it is trained jointly with both modalities and the modality-dependent features are not compatible, it is encouraged to rely on modality-invariant features. The result that modality-agnostic decoders are outperforming modality-specific decoders trained on captions for decoding captions confirms this, because if the decoder was only relying on modality-dependent features the addition of additional training data from another stimulus modality could not increase the performance. (Also, the lack of a performance drop compared to modality-specific decoders trained on images is only possible thanks to the reliance on modality-invariant features. If the decoder only relied on modality-dependent features the addition of data from another modality would equal an addition of noise to the training data which must result in a performance drop at test time.). We can not exclude the possibility that modality-agnostic decoders are also relying on modality-dependent features, but our results suggest that they are relying at least to some degree on modality-invariant features.

Second, the authors claim that "modality-specific decoders can be applied only in the modality that they were trained on, while "modality-agnostic decoders can be applied to decode stimuli from multiple modalities, even without knowing a priori the modality the stimulus was presented in" (line 47). While "modality-agnostic" decoders do outperform "modality-specific" decoders in the cross-modality conditions, it is important to note that "modality-specific" decoders still perform better than expected by chance (figure 5). It is also important to note that knowing about the input modality still improves decoding performance even for "modality-agnostic" decoders, since it determines the optimal feature space-it is better to decode brain responses to images using decoders trained on image features, and it is better to decode brain responses to captions using decoders trained on caption features.

Thanks for this important remark. We corrected this statement and now say that “modality-specific decoders that are trained to be applied only in the modality that they were trained on”, highlighting that their training process optimizes them for decoding in a specific modality. They can indeed be applied to the other modality at test time, this however results in a substantial performance drop.

It is true that knowing the input modality can improve performance even for modality-agnostic decoders. This can most likely be explained by the fact that in that case the decoder can leverage both, modality-invariant and modality-dependent features. We will not further focus on this result however as the main motivation to build modality-agnostic decoders is to be able to decode stimuli without knowing the stimulus modality a priori.

**Recommendations for the authors:**

**Reviewer #1 (Recommendations for the authors):**
I will list additional recommendations below in no specific order:(1) I find the term "modality agnostic" quite unusual, and I believe I haven't seen it used outside of the ML community. I would urge the authors to change the terminology to be more common, or at least very early explain why the term is much better suited than the range of existing terms. A modality agnostic representation implies that it is not committed to a specific modality, but it seems that a representation cannot be committed to something.

In the updated manuscript we now refer to the identified brain patterns as modality-invariant, which has previously been used in the literature (Man et al. 2012; Devereux et al. 2013; Patterson et al. 2016; Deniz et al. 2019, Nakai et al. 2021) (see also the general response on top and the Introduction and Related Work sections of the updated manuscript).

We continue to refer to the decoders as modality-agnostic, as this is a new type of decoder, and describes the fact that they are trained in a way that abstracts away from the modality of the stimuli. We chose this term as we are not aware of any work in which brain decoders were trained jointly on multiple stimulus modalities and in order not to risk contradictions/confusions with other definitions.

Deniz, F., Nunez-Elizalde, A. O., Huth, A. G., & Gallant, J. L. (2019). The Representation of Semantic Information Across Human Cerebral Cortex During Listening Versus Reading Is Invariant to Stimulus Modality. Journal of Neuroscience, 39(39), 7722–7736. https://doi.org/10.1523/JNEUROSCI.0675-19.2019

Devereux, B. J., Clarke, A., Marouchos, A., & Tyler, L. K. (2013). Representational Similarity Analysis Reveals Commonalities and Differences in the Semantic Processing of Words and Objects. The Journal of Neuroscience, 33(48).

Nakai, T., Yamaguchi, H. Q., & Nishimoto, S. (2021). Convergence of Modality Invariance and Attention Selectivity in the Cortical Semantic Circuit. Cerebral Cortex, 31(10), 4825–4839. https://doi.org/10.1093/cercor/bhab125

Man, K., Kaplan, J. T., Damasio, A., & Meyer, K. (2012). Sight and Sound Converge to Form Modality-Invariant Representations in Temporoparietal Cortex. Journal of Neuroscience, 32(47), 16629–16636.

Patterson, K., & Lambon Ralph, M. A. (2016). The Hub-and-Spoke Hypothesis of Semantic Memory. In Neurobiology of Language (pp. 765–775). Elsevier. https://doi.org/10.1016/B978-0-12-407794-2.00061-4

(2) The table in Figure 1B would benefit from also highlighting the number of stimuli that have overlapping captions and images.

The number of overlapping stimuli is rather small (153-211 stimuli depending on the subject). We added this information to Table 1B.

(3) The authors wrote that training stimuli were presented only once, yet they used a one-back task. Did the authors also exclude the first presentation of these stimuli?

Thanks for pointing this out. It is indeed true that some training stimuli were presented more than once, but only for the case of one-back target trials. In these cases the second presentation of the stimulus was excluded, but not the first. As the subject can not be aware of the fact that the upcoming presentation is going to be a one-back target, the first presentation can not be affected by the presence of the subsequent repeated presentation. We updated the manuscript to clarify this issue.

(4) Coco has roughly 80-90 categories, so many image captions will be extremely similar (e.g., "a giraffe walking", "a surfer on a wave", etc.). How can people keep these apart?

It is true that some captions and images are highly similar even though they are not matching in the dataset. This might result in several false button presses because the subjects identified an image-caption pair as matching when in fact it wasn't intended to. However, as there was no feedback given on the task performance, this issue should not have had a major influence on the brain activity of the participants.

(5) Footnotes for statistics are quite unusual - could the authors integrate statistics into the text?

Thanks for this remark, in the updated manuscript all statistics are part of the main text.

(6) It may be difficult to achieve the assumptions of a permutation test - exchangeability, which may bias statistical results. It is not uncommon for densely sampled datasets to use bootstrap sampling on the predictions of the test data to identify if a given percentile of that distribution crosses 0. The lowest p-value is given by the number of bootstrap samples (e.g., if all 10,000 bootstrap samples are above chance, then p < 0.0001). This may turn out to be more effective.

Thanks for this comment. Our statistical procedure was in fact involving a bootstrapping procedure to generate a null distribution on the group-level. We updated the manuscript to describe this method in more detail. Here is the updated paragraph: “To estimate the statistical significance of the resulting clusters we performed a permutation test, combined with a bootstrapping procedure to estimate a group-level null distribution see also Stelzer et al., 2013. For each subject, we evaluated the decoders 100 times with shuffled labels to create per-subject chance-level results. Then, we randomly selected one of the 100 chance-level results for each of the 6 subjects and calculated group-level statistics (TFCE values) the exact same way as described in the preceding paragraph. We repeated this procedure 10,000 times resulting in 10,000 permuted group-level results. We ensured that every permutation was unique, i.e. no two permutations were based on the same combination of selected chance-level results. Based on this null distribution, we calculated p-values for each vertex by calculating the proportion of sampled permutations where the TFCE value was greater than the observed TFCE value. To control for multiple comparisons across space, we always considered the maximum TFCE score across vertices for each group-level permutation (Smith and Nichols, 2009).”

(7) The authors present no statistical evidence for some of their claims (e.g., lines 335-337). It would be good if they could complement this in their description. Further, the visualization in Figure 4 is rather opaque. It would help if the authors could add a separate bar for the average modality-specific and modality-agnostic decoders or present results in a scatter plot, showing modality-specific on the x-axis and modality-agnostic on the y-axis and color-code the modality (i.e., making it two scatter colors, one for images, one for captions). All points will end up above the diagonal.

We updated the manuscript and added statistical evidence for the claims made:

We now report results for the claim that when considering the average decoding performance for images and captions, modality-agnostic decoders perform better than modality-specific decoders, irrespective of the features that the decoders were trained on.

Additionally, we report the average modality-agnostic and modality-specific decoding accuracies corresponding to Figure 4. For modality-agnostic decoders the average value is 81.86\%, for modality-specific decoders trained on images 78.15\%, and for modality-specific decoders trained on captions 72.52\%. We did not add a separate bar to Figure 4 as this would add additional information to a Figure which is already very dense in its information content (cf. Reviewers 2’s recommendations for the authors). We therefore believe it is more useful to report the average values in the text and provide results for a statistical test comparing the decoder types. A scatter plot would make it difficult to include detailed information on the features, which we believe is crucial.

We further provide statistical evidence for the observation regarding the directionality of cross-modal decoding.

**Reviewer #2 (Recommendations for the authors):**
For achieving more evidence to support modality-agnostic representations in the brain, I suggest more thorough analyses, for example:(1) Traditional searchlight RSA using different deep learning models. Through this approach, it might identify different brain areas that are sensitive to different formats of information (visual, text, multimodal); subsequently, compare the decoding performance using these ROIs.(2) Build more dissociable decoders for information of different modality formats, if possible. While I do not have a concrete proposal, more targeted decoder designs might better dissociate representational formats (i.e., unimodal vs. modality-agnostic).(3) A more detailed exploration of the "qualitative decoding results"--for example, quantitatively examining error types produced by modality-agnostic versus modality-specific decoders--would be informative for clarifying what specific content the decoder captures, potentially providing stronger evidence for modality-agnostic representations.

Thanks for these suggestions. As the main goal of the paper is to introduce modality-agnostic decoders (which should be more clear from the updated manuscript, see also the general response to reviews), we did not include alternative methods for identifying modality-invariant regions. Nonetheless, we agree that in order to obtain more in-depth insight into the nature of representations that were recorded, performing analyses with additional methods such as RSA, comparisons with more targeted decoder designs in terms of their target features will be indispensable, as well as more in-depth error type analyses. We leave these analyses as promising directions for future work.

The writing could be further improved in the introduction and, accordingly, the discussion. The authors listed a series of theories about conceptual representations; however, they did not systematically explain the relationships and controversies between them, and it seems that they did not aim to address the issues raised by these theories anyway. Thus, the extraction of core ideas is suggested. The difference between "modality-agnostic" and terms like "modality-independent," "modality-invariant," "abstract," "amodal," or "supramodal," and the necessity for a novel term should be articulated.

The updated manuscript includes an improved introduction and discussion section that highlight the main focus and contributions of the study.

We believe that a systematic comparison of theories on conceptual representations involving their relationships and controversies would require a dedicated review paper. Here, we focused on the aspects that are relevant for the study at hand (modality-invariant representations), for which we find that none of the considered theories can be rejected based on our results.

Regarding the terminology (modality-agnostic vs. modality-invariant, ..) please refer to the general response.

The figures also have room to improve. For example, Figures 4 and 5 present dense bar plots comparing multiple decoding settings (e.g., modality-specific vs. modality-agnostic decoders, feature space, within-modal vs. cross-modal, etc.); while comprehensive, they would benefit from clearer labels or separated subplots to aid interpretation. All figures are recommended to be optimized for greater clarity and directness in future revisions.

Thanks for this remark. We agree that the figures are quite dense in information. However, splitting them up into subplots (e.g. separate subplots for different decoder types) would make it much less straightforward to compare the accuracy scores between conditions. As the main goal of these figures is to compare features and decoder types, we believe that it is useful to keep all information in the same plot.

You are also suggesting to improve the clarity of the labels. It is true that the top left legend of Figures 4 and 5 was mixing information about decoder type and broad classes of features (vision/language/multimodal). To improve clarity, we updated the figures and clearly separated information on decoder type (the hue of different bars) and features (x-axis labels). The broad classes of features (vision/language/multimodal) are distinguished by alternating light gray background colors and additional labels at the very bottom of the plots.

The new plots allow for easy performance comparison of the different decoder types and additionally provide information on confidence intervals for the performance of modality-specific decoders, which was not available in the previous figures.

**Reviewer #3 (Recommendations for the authors):**
(1) As discussed in the Public Review, I think the paper would greatly benefit from clearer terminology. Instead of describing the decoders as "modality-agnostic" and "modality-specific", perhaps the authors could describe the decoding conditions based on the train and test modalities (e.g., "image-to-image", "caption-to-image", "multimodal-to-image") or using the terminology from Figure 3 (e.g., "within-modality", "cross-modality", "modality-agnostic").

We updated our terminology to be clearer and more accurate, as outlined in the general response. The terms modality-agnostic and modality-specific refer to the training conditions, and the test conditions are described in Figure 3 and are used throughout the paper.

(2) Line 244: I think the multimodal one-back task is an important aspect of the dataset that is worth highlighting. It seems to be a relatively novel paradigm, and it might help ensure that the participants are activating modality-agnostic representations.

It is true that the multimodal one-back task could play an important role for the activation of modality-invariant representations. Future work could investigate to what degree the presence of widespread modality-invariant representations is dependent on such a paradigm.

(3) Line 253: Could the authors elaborate on why they chose a random set of training stimuli for each participant? Is it to make the searchlight analyses more robust?

A random set of training stimuli was chosen in order to maximize the diversity of the training sets, i.e. to avoid bias based on a specific subsample of the CoCo dataset. Between-subject comparisons can still be made based on the test set which was shared for all subjects, with the limitation that performance differences due to individual differences or to the different training sets can not be disentangled. However, the main goal of the data collection was not to make between-subject comparisons based on common training sets, but rather to make group-level analyses based on a large and maximally diverse dataset.

(4) Figure 4: Could the authors comment more on the patterns of decoding performance in Figure 5? For instance, it is interesting that ResNet is a better target than ViT, and BERT-base is a better target than BERT-large.

A multitude of factors influence the decoding performance, such as features dimensionality, model architecture, training data, and training objective(s) (Conwell et al. 2023; Raugel et al. 2025). Bert-base might be better than bert-large because the extracted features are of lower dimension. Resnet might be better than ViT because of its architecture (CNN vs. Transformer). To dive deeper into these differences further controlled analysis would be necessary, but this is not the focus of this paper. The main objective of the feature comparison was to provide a broad overview over visual/linguistic/multimodal feature spaces and to identify the most suitable features for modality-agnostic decoding.

Conwell, C., Prince, J. S., Kay, K. N., Alvarez, G. A., & Konkle, T. (2023). What can 1.8 billion regressions tell us about the pressures shaping high-level visual representation in brains and machines? (p. 2022.03.28.485868). bioRxiv. https://doi.org/10.1101/2022.03.28.485868

Raugel, J., Szafraniec, M., Vo, H.V., Couprie, C., Labatut, P., Bojanowski, P., Wyart, V. and King, J.R. (2025). Disentangling the Factors of Convergence between Brains and Computer Vision Models. arXiv preprint arXiv:2508.18226.

(5) Figure 7: It is interesting that the modality-agnostic decoder predictions mostly appear traffic-related. Is there a possibility that the model always produces traffic-related predictions, making it trivially correct for the presented stimuli that are actually traffic-related? It could be helpful to include some examples where the decoder produces other types of predictions to dispel this concern.

The presented qualitative examples were randomly selected. To make sure that the decoder is not always predicting traffic-related content, we included 5 additional randomly selected examples in Figures 6 and 7 of the updated manuscript. In only one of the 5 new examples the decoder was predicting traffic-related content, and in this case the stimulus had actually been traffic-related (a bus).